# Orderly mitosis shapes interphase genome architecture

**Krishnendu Guin[1], Adib Keikhosravi[2], Raj Chari[3], Gianluca Pegoraro[2], Tom Misteli[1]\***

[1]National Cancer Institute, NIH, Bethesda, United States; [2]High Throughput Imaging Facility (HiTIF), National Cancer Institute, NIH, Bethesda, United States; [3]Genome Modification Core (GMC), Frederick National Lab for Cancer Research, Frederick, United States

## eLife Assessment

This **important** study combines microscopy and CRISPR screening to identify factors involved in global chromatin organisation, using centromere clustering as a proxy. The authors present **solid** evidence demonstrating that acute depletion of a range of mitotic regulators alters centromere distribution in interphase. The work will be of interest to researchers studying genome organisation, nuclear architecture, chromosome biology, and the mechanisms linking mitosis to interphase nuclear organisation.

**\*For correspondence:**
mistelit@mail.nih.gov

**Competing interest:** The authors declare that no competing interests exist.

**Abstract** Genomes assume a complex 3D architecture in the interphase cell nucleus. Yet the molecular mechanisms that determine global genome architecture are only poorly understood. To identify mechanisms of higher-order genome organization, we performed high-throughput imaging-based CRISPR knockout screens targeting 1064 genes encoding nuclear proteins in multiple human cell lines. We assessed changes in the distribution of centromeres at single-cell resolution as surrogate markers for global genome organization. The screens revealed multiple major regulators of spatial distribution of centromeres, including components of the nucleolus, kinetochore, cohesins, condensins, and the nuclear pore complex. Alterations in centromere distribution required progression through the cell cycle and acute depletion of mitotic factors with distinct functions altered centromere distribution in the subsequent interphase. These results identify molecular determinants of spatial centromere organization, and they show that orderly progression through mitosis shapes interphase genome architecture.

## Introduction

Genomes are complex polymers. In the cell nucleus, the genome is organized via several structures at different length scales. At the shortest scale, genomic DNA is wrapped around a histone octamer to form a nucleosome (*Wen et al., 2025*). Strings of nucleosomes then fold onto themselves to form a chromatin fiber, which in turn organizes itself into chromatin loops, typically in the 1–100 kb range. Chromatin fibers further fold into 0.5–2 Mb topologically associated domains (TADs), which can homotypically associate with each other into transcriptionally active A compartments and transcriptionally repressed B compartments spanning several megabases (*Rowley and Corces, 2018*; *Hildebrand and Dekker, 2020*; *Misteli, 2020*; *Paldi and Cavalli, 2026*). While these genome features occur in most cell types and species, all chromatin features also exhibit extensive single-cell variability (*Finn and Misteli, 2019*).

The organization of genomes is non-random within the cell nucleus (*Parada and Misteli, 2002*; *Oliver and Misteli, 2005*; *Bouwman et al., 2022*). Chromosomes and individual gene loci tend to occupy preferred positions relative to the nuclear boundary and relative to each other (*Shachar and Misteli, 2017*; *Scholz et al., 2019*; *Bouwman et al., 2022*). For example, the chromosomes that contain clusters of ribosomal genes congregate in 3D space to form the subnuclear compartment of the nucleolus (*Dundr et al., 2000*). Similarly, transcriptionally repressive genome regions are often associated with the nuclear lamina at the periphery of the cell nucleus and around the nucleolus (*Croft et al., 1999*; *Alagna et al., 2023*). Defects in spatial genome organization are associated with multiple diseases, including cancer and accelerated aging (*Misteli, 2010*; *Wang et al., 2023*; *Amodeo et al., 2025*).

Recent studies have shed light on the mechanisms that determine the local organization of the genome (*Bonev and Cavalli, 2016*; *Finn and Misteli, 2019*; *Misteli, 2020*). Chromatin loops and domains are formed by loop extrusion, in which the ring-like condensin protein complex acts as a molecular motor (*Terakawa et al., 2017*) to extrude the chromatin fiber to form loops, which can eventually congregate into TADs (*Ganji et al., 2018*; *Davidson and Peters, 2021*). In contrast, the molecular mechanisms that determine the higher-order global genome organization, such as the location of genes, chromatin domains, or chromosomes within the 3D space of the nucleus, are less clear. Some insights come from the observation in yeast and *Caenorhabditis elegans* demonstrating that transcriptionally repressive chromosomes are preferentially tethered to the nuclear periphery via histone modifications (*Towbin et al., 2012*; *Shachar and Misteli, 2017*). Furthermore, unbiased screening approaches have suggested that progression through S-phase is essential for establishing the nuclear location of individual genes (*Joyce et al., 2012*; *Shachar et al., 2015*). It has also been suggested that the propensity to undergo homotypic interactions promotes the clustering of similar genome regions, e.g., the association of ribosomal genes in the nucleolus (*Lafontaine et al., 2021*) or the formation of intranuclear heterochromatin blocks (*Misteli, 2020*).

The centromere is a prominent structural feature of all chromosomes (*Murray and Szostak, 1985*). Centromeres are specialized genomic loci that assemble the kinetochore protein complex, which connects chromosomes with the microtubule spindle during mitosis, and through their attachment ensure error-free chromosome segregation (*McKinley and Cheeseman, 2016*). Like other chromosomal features, centromeres have been observed to assume non-random locations in the cell nucleus across species. In yeast, centromeres cluster and localize at the nuclear periphery at some or all stages of the cell cycle (*Guin et al., 2020*). Variable degrees of clustering have been observed in apicomplexan parasites (*Bunnik et al., 2019*), plants (*Fransz et al., 2002*), flies (*Padeken et al., 2013*), and mice (*Weierich et al., 2003*; *Stevens et al., 2017*), where peri-centromeres cluster into prominent chromocenters, presumably via homotypic interactions (*Brändle et al., 2022*). In humans, centromere clustering is less pronounced, but increased clustering of centromeres near nucleoli has been observed in multiple cell lines (*Weierich et al., 2003*; *Bury et al., 2020*; *Rodrigues et al., 2023*; *Kumar et al., 2024*), particularly prominently in human stem cells where most centromeres are localized near nucleoli (*Wiblin et al., 2005*; *Rodrigues et al., 2023*). The fact that clustered centromeres tend to dissociate from the nucleolus during stem cell differentiation (*Rodrigues et al., 2023*) may point to a functional role of nucleolar centromere clustering. However, the underlying molecular mechanisms determining spatial centromere distribution remain elusive.

Given their prominent nature and non-random location in the cell nucleus, we used centromeres as proxies for higher-order spatial genome organization to identify molecular determinants of global genome architecture. We tested 1064 chromatin-associated proteins in high-throughput imaging (HTI)-based CRISPR/Cas9 knockout (KO) screens in human cell lines to identify conserved molecular determinants of nuclear centromere distribution. Our data identifies proteins implicated in diverse biological functions. By impairing the function of several of these candidates during the cell cycle, we demonstrate that defective mitotic progression alters centromere distribution in the daughter cells. We conclude that orderly progression through mitosis shapes global 3D genome architecture.

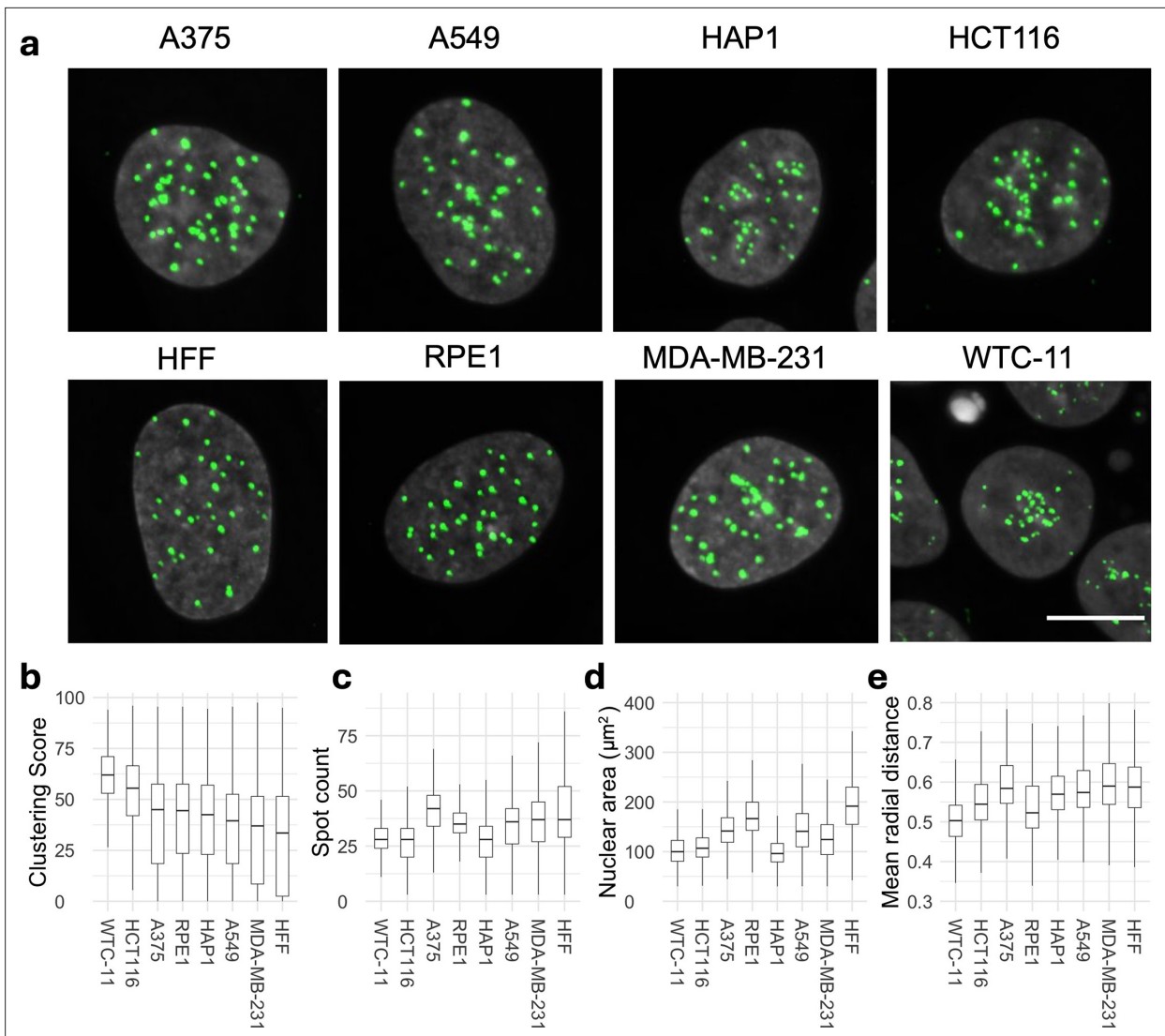

**Figure 1.** Spatial organization of centromeres is cell-type specific in human cell lines. (**a**) Representative images of CENP-C (green) and DAPI (gray) stained nuclei in indicated human cell lines. Scale bar: 10 μm. (**b, c**) Spatial organization of centromeres quantified using Ripley K's clustering score (**b**), CENP-C spot count (**c**). (**d**) Nuclear area and (**e**) mean radial distance in human cell lines. Statistical significance of differences between cell lines for clustering score, spot count, mean radial distance, and nuclear area was tested using analysis of variance (ANOVA) (p-value or 'Pr(>F)'<2e-16) following Tukey's HSD test to compare means of all pairs of cell lines. Box plots represent the interquartile range (IQR) between the first and third quartiles (box), the median (horizontal bar), and the whiskers that extend to the highest and lowest data points within 1.5 times the IQR. Values are from one representative experiment with at least 7 technical replicates. At least 1000 cells were analyzed in each category per experiment.

The online version of this article includes the following figure supplement(s) for figure 1:

**Figure supplement 1.** Quantification of centromere clustering using CENP-A and CENP-C as centromere markers.

## Results

### The spatial distribution of centromeres is cell-type specific

We first sought to quantitatively profile the spatial distribution of centromeres in human cells (**Figure 1**). We used HTI to visualize endogenous centromeres in eight human cell lines from different tissues and with distinct proliferation properties, including immortalized retinal pigment epithelium RPE1 cells, immortalized human HFF fibroblasts, the induced pluripotent stem cell (iPSC) line WTC-11, and several cancer cell lines of different origin (**Supplementary file 1**). Some of these cell lines contain numerical aberrations of chromosomes (**Giard et al., 1973**; **Kotecki et al., 1999**), which were taken into account when setting the baseline for the quantitative analysis of centromere distribution

in individual cell lines. Centromeres were visualized by indirect immunofluorescence (IF) for the integral kinetochore component CENP-C, which localizes to centromeres at all stages of the cell cycle and completely colocalized throughout the cell cycle with the centromere protein CENP-A (*Hori et al., 2008*; *Klare et al., 2015*; *Figure 1—figure supplement 1a*). CENP-C was used as a marker for centromeres in all subsequent experiments.

We quantified the number of centromere spots per nucleus and analyzed centromere spatial distribution in the nucleus by HTI in several thousand cells per cell line by using HiTIPS, an open-source HTI analysis platform that accurately segments nuclei and centromeres in large HTI image datasets (*Figure 1—figure supplement 1b*; *Keikhosravi et al., 2024*). For each cell, we measured the number of centromeres per nucleus as spot count and also derived a centromere clustering score, which measures the overall distribution of centromeres in the nucleus (*Figure 1—figure supplement 1c and d*, also see Methods). The clustering score is a metric derived from the Ripley's K function, which we established in pilot experiments as a robust and sensitive measure of centromere clustering (*Keikhosravi et al., 2025a*). The clustering score quantifies deviations of the centromere distribution from uniformly distributed spots and it is normalized to nuclear size. Importantly, the clustering score is robust to changes in the centromere spot number and thus accounts for any differences in centromere numbers in the various cell lines or due to aneuploidy (*Keikhosravi et al., 2025b*).

We observed significant qualitative differences in centromere distribution among human cell lines (*Figure 1a*). For example, in WTC-11 cells, centromeres were strongly clustered, in line with centromere association with the nucleolus observed in other human stem cells (*Wiblin et al., 2005*; *Rodrigues et al., 2023*). In contrast, A549 basal epithelial cells derived from lung cancer and MDA-MB-231 epithelial-like breast cancer cells showed noticeably less clustering than HFFs, which exhibited the most dispersed distribution among the cell lines tested (*Figure 1a*). These visual trends were confirmed by quantitative HTI analysis using the clustering score, which was highest for WTC-11 cells and lowest for HFFs (*Figure 1b*). These differences in clustering were unrelated to spot number or to nuclear area (*Figure 1c and d*). In addition, WTC-11 cells had the lowest and HFF cells the highest median population values for the mean normalized radial CENP-C distance, which represents the per-cell average distance of centromeres from the center of the nucleus (*Figure 1e*), consistent with the differential clustering behavior in these two cell lines. Statistical analysis of variance (ANOVA) indicated that most cell lines were significantly different from each other based on centromere clustering score or spot count (*Supplementary file 2*), indicating cell-type specificity of centromere distribution patterns. We also noted cell-to-cell variation for all centromere distribution parameters within the population (*Figure 1b, c, and e*), demonstrating single-cell heterogeneity of centromere distributions as previously observed for various other features of genome organization (*Finn and Misteli, 2019*). In conclusion, quantitative HTI analysis of centromere localizations in thousands of single cells shows that spatial patterns of centromeres in the human cell nucleus are cell-type specific.

## Imaging-based CRISPR-KO screens identify regulators of centromere clustering

Having established the heterogeneous and non-random nature of centromere clustering in the nucleus, we sought to identify the molecular basis for this phenomenon. To do so, we developed an arrayed HTI-based CRISPR-KO screening assay to identify regulators of the spatial distribution of centromeres (*Figure 2a*). For the screens, we designed an sgRNA library targeting 1064 genes encoding nuclear proteins, enriched in structural components of the nucleus, epigenetic modifiers, and components of the genome maintenance and expression machinery (for library composition, see *Supplementary file 3*). A non-targeting, scrambled sgRNA and sgRNAs targeting the non-expressing *OR10A5* gene were used as negative controls for sgRNA transfection and for CRISPR-induced DNA damage response (*Liu et al., 2019*), respectively. In addition, as a positive control for sgRNAs transfection, we used sgRNAs against the essential *PLK1* gene whose ablation results in rapid and extensive cell death (*Schibler et al., 2023*). As a positive control, we used sgRNAs targeting the condensin II complex component *NCAPH2*, whose silencing has previously been shown to induce clustering of centromeres (*Hoencamp et al., 2021*). In light of our observation that spatial patterns of centromere distribution can be different between cell lines, we performed screens in two cell lines, HCT116 and RPE1, which represent clustered and unclustered centromere patterns, respectively (see *Figure 1a*). For quantitative HTI analysis, we performed imaging-based phenotypic scoring of centromere distribution patterns using

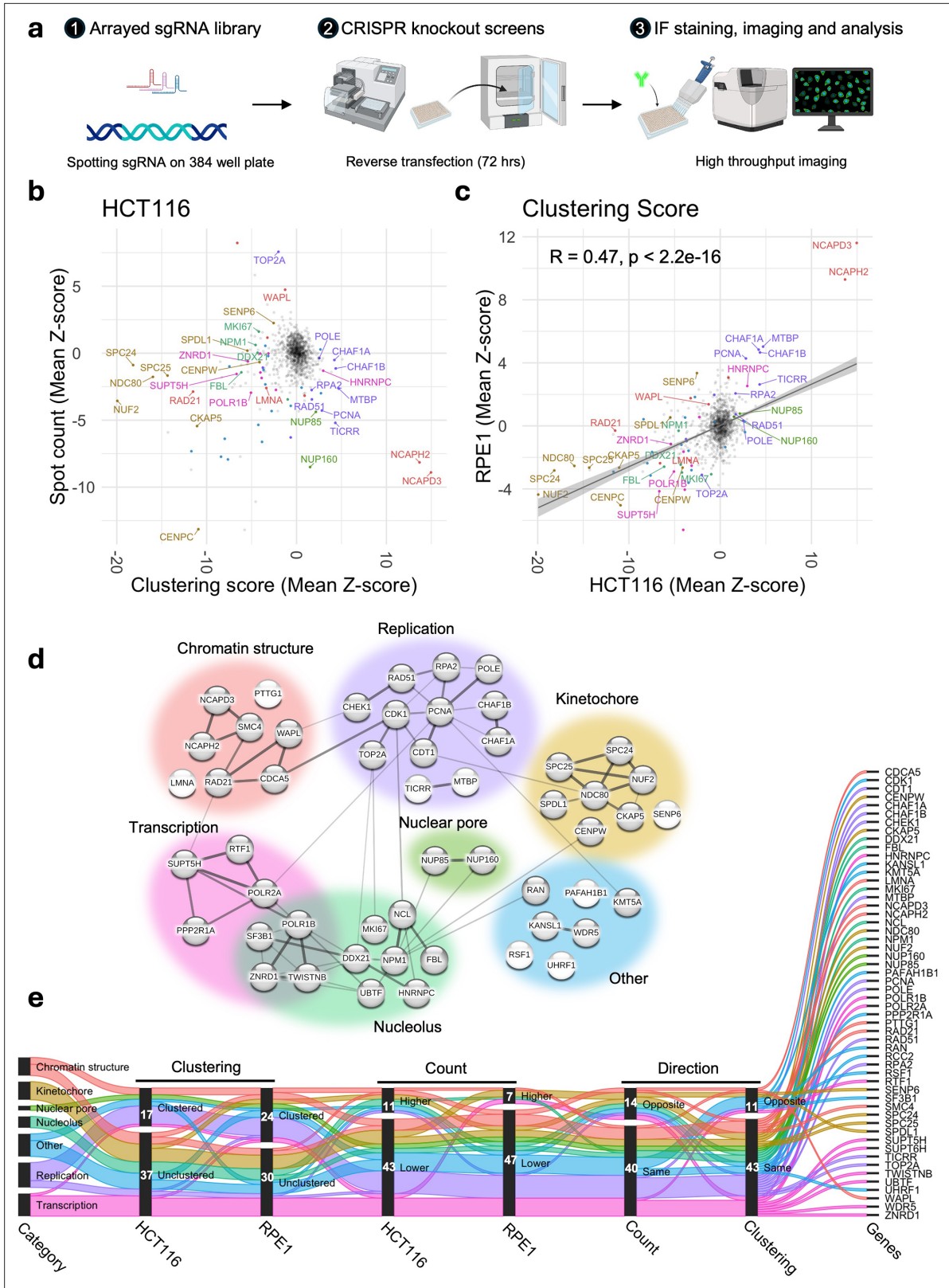

**Figure 2.** Identification of the molecular determinants of spatial centromere distribution across cell types. (**a**) Schematics showing three stages of high-throughput imaging-based arrayed CRISPR knockout screen employed to identify molecular determinants of spatial centromere distribution. (**b**) Changes in spot count (mean Z-score of two replicates, y-axis) and clustering score (mean Z-score of two replicates, x-axis) for each of the 1064 sgRNAs. The most prominent hits were labeled and color-coded as in **d**. Non-hits are colored in gray. (**c**) Changes in clustering score in HCT116 (mean Z-score

*Figure 2 continued on next page*

*Figure 2 continued*

of two replicates, x-axis) and in RPE1 (mean Z-score of two replicates, y-axis) cells for each of the 1064 sgRNAs. Hits and non-hits are color-coded and labeled as in **d**. A linear trend line (gray) was fitted to the data, and Pearson's correlation coefficient calculated is indicated at the top-left corner of the plot. (**d**) Network diagram with lines between 52 common hits drawn based on known physical and/or genetic interactions generated by the STRING database. The thickness of the lines indicates higher strength of data supporting the interaction. Broad categories are color-coded as indicated. (**e**) Plot shows changes in clustering (clustered or unclustered), count (higher or lower), and direction between two cell lines (same or opposite) for each of the common genes that are color-coded based on their category as in **d**. Counts of genes in each subcategory are indicated. Values represent two biological replicates. Typically, 200–500 cells were analyzed for each target gene per experiment.

The online version of this article includes the following figure supplement(s) for figure 2:

**Figure supplement 1.** CRISPR knockout screens for centromere distribution phenotypes in HCT116 cells are reproducible.

**Figure supplement 2.** Identification of the molecular determinants of spatial centromere distribution in RPE1 cells.

**Figure supplement 3.** Validation of screen hits, cell-cycle analysis of clustering factor knockdown, and their effect on clustering score.

**Figure supplement 4.** Comparative analysis of cell lines confirms common molecular determinants of spatial centromere distribution.

centromere spot count and the Ripley's K-based clustering score as read-out parameters (*Keikhosravi et al., 2025a*). All CRISPR-KO screens were performed in biological duplicates and generated data from a few hundred to over a thousand cells per replicate for each target gene (*Supplementary files 4 and 5*).

The results of the screens indicated consistent phenotypic separation of the positive and negative controls (*Figure 2—figure supplement 1a and b*) and high reproducibility of hits in the two biological replicates for both cell lines (*Figure 2—figure supplement 1c and d*). We defined hits as sgRNA perturbations that altered either spot count or clustering by a Z-score of at least 2.5 units from the median phenotype of all sgRNAs included in the library (*Supplementary file 3*). Any sgRNA KOs that resulted in a large number of dismorphic and/or abnormally sized nuclei, as measured by nucleus compactness and/or nuclear area, were filtered out from subsequent steps of the analysis. We excluded from analysis sgRNAs which resulted in high cytotoxicity (cell number Z-score <–2.5), or which produced inconsistent results across the two biological replicates (see Methods).

Following these criteria, we identified 111 genes whose CRISPR-KO altered centromere distribution in HCT116 cells (*Figure 2b*, *Supplementary file 4*). Among these, 80% (89/111) altered the CENP-C clustering score, 41% (45/111) altered CENP-C spot count, and 20% (23/111) altered both parameters. The majority of hits (81%; 72/89) unclustered centromeres, whereas 19% (17/89) increased clustering (*Supplementary file 4*). Among the 23 genes that altered both parameters, six increased the clustering score and decreased spot count, indicating higher clustering, whereas the opposite trend was observed for four genes, indicating dispersion (*Supplementary file 4*). The remaining genes (13/23) concomitantly decreased the clustering score and spot count, suggesting global dispersion but local clustering of centromeres into fewer but larger local clusters (*Supplementary file 4*). A concomitant increase in both spot count and clustering score was not observed. The effects on centromere distribution did not correlate with changes in nuclear area (*Figure 2—figure supplement 1e and f*).

We similarly identified 113 hits when we performed the CRISPR-KO screen in RPE1 cells, which are characterized by a lower degree of centromere clustering than HCT116 cells (*Figure 2—figure supplement 2*, *Supplementary file 5*). Similar to HCT116 cells, we observed a nonlinear relationship of spot count and clustering score in RPE1 cells (*Supplementary file 5*). The majority (77%, 87/113) of identified sgRNAs altered centromere spot count, 40% altered the clustering score, while 17% altered both parameters (*Figure 2c*, *Supplementary file 5*). When analyzed using the clustering score, the majority (58%, 26/45) of hits dispersed centromeres, whereas the rest (42%, 19/45) increased clustering (*Supplementary file 5*).

Select hits were orthogonally validated using siRNA knockdown with a validation rate of 90% (27/30) (*Figure 2—figure supplement 3a*). Reassuringly, in line with known centromere-nucleoli association (*Bury et al., 2020*), several nucleolar proteins, including NPM1, NCL, and FBL, were identified as hits, confirming the validity of our screening approach. In addition, our positive control NCAPH2, represented in the library, and another condensin II component NCAPD3 were strong hits in both cell lines and in all replicates of the screen. We identified both essential and non-essential genes, and only

very few hits altered cell-cycle distribution, indicating that the hits were not due to secondary effects on the cell cycle or cell viability (*Figure 2—figure supplement 3b and c*). In addition, centromere clustering levels were generally similar in G1, S, and G2/M phases for most hits compared to scrambled control, except for a handful of cases where the pattern changed upon knockdown of target genes (*Figure 2—figure supplement 3d*).

A comparative analysis of the CRISPR-KO screen results in HCT116 and in RPE1 cells indicated that KO of most genes similarly altered centromere distributions in both cell lines, but that the extent of change (Z-score) could vary depending on the initial state of centromere distribution (R=0.47, p<10$^{-10}$, *Figure 2c*). We identified 52 genes that alter centromere distribution in both cell lines (*Figure 2—figure supplement 4a*). The majority of these genes altered phenotypes in the same direction for clustering score (79%, 41/52) and spot count (73%, 38/52) (*Figure 2e*). Only rare examples of cell-type-specific opposite effects were observed (*Figure 2e* and *Figure 2—figure supplement 4b*). Similarly, we identified genes that altered centromere distribution in only one cell line (*Figure 2—figure supplement 4*, *Supplementary files 4 and 5*). Taken together, these data identify both conserved and cell-type-specific regulators of centromere distribution.

To gain insights into the functions of the common hits, we used STRING analysis which identifies pathways based on known physical and genetic interactions (*Figure 2d*; *Szklarczyk et al., 2025*). Based on this analysis, centromere distribution modifiers were grouped into six categories: regulators of chromatin structure, kinetochore proteins, nucleolar proteins, nuclear pore complex components, replication factors, and transcription-associated factors (*Figure 2d*). Interestingly, while KO of most replication- and nuclear pore-associated genes increased clustering, KO of kinetochore components and transcription-associated factors led predominantly to centromere dispersion in both cell types (*Figure 2e* and *Figure 2—figure supplement 4b*). Loss of proteins implicated in chromatin structure or the nucleolus either clustered or dispersed centromeres in a gene-specific manner (*Figure 2e* and *Figure 2—figure supplement 4b*). It is also important to note that factors grouped into these broad classes may perform functions in multiple categories. For example, the NUP107-160 complex, which is a prominent structural component of the nuclear pore, is known to also interact with kinetochores (*Orjalo et al., 2006*). Taken together, these results identify major regulators of spatial centromere organization.

## Spatial redistribution of centromeres requires cell-cycle progression

The identified regulators of centromere distribution are involved in diverse cellular functions and pathways, suggesting multilayered control of centromere distribution. To gain mechanistic insight, we first asked whether the identified regulators act at particular points in the cell cycle. To establish a baseline for analysis, we quantitated centromere distribution in G1, S, and G2/M of cell-cycle staged HCT116 and RPE1 cells based on DAPI and EdU pulse labeling as described before (*Salic and Mitchison, 2008*; *Bruhn et al., 2014*; *Roukos et al., 2015*; *Figure 3a and b*). As expected, due to the duplication of the genome during replication, the number of detectable centromere spots increased in S-phase cells (p=0.009) and was highest in G2/M HCT116 cells (p<10$^{-10}$; *Figure 3c*). A marginal increase in clustering score was observed in G2/M cells compared to G1 cells (*Figure 3d*; p=0.002), while radial positioning of centromeres remained mostly unchanged except for a small increase in G1 cells (*Figure 3e*; p=0.004). A similar trend was observed in RPE1 cells for all three parameters (*Figure 3c, d, and e*). As expected, the nuclear area was significantly increased in S and G2/M in both HCT116 and RPE1 cells (*Figure 3f*, p<10$^{-10}$). The lack of strong correlation between nuclear area and clustering score within G1, S, or G2/M subpopulations (R<0.3) indicates that increased clustering scores in G2/M cells are unrelated to nuclear size increase (*Figure 3g*). We conclude that, in line with observations on radial position of genomic loci (*Shachar et al., 2015*) and of chromosome territories (*Jowhar et al., 2018*), the overall distribution of centromeres does not vary strongly within the interphase of the cell cycle.

To specifically ask whether the identified hits required progression through the cell cycle, we performed siRNA knockdown of 30 select hits in asynchronous cells or in cells that were either arrested at the G1/S boundary by standard double thymidine block (*Chen and Deng, 2018*) or at the G2/M boundary by treatment with the CDK1 inhibitor RO-3306 as previously described (*Vassilev et al., 2006*; *Figure 3g*; see Methods). Loss of cell viability upon transfection of siDeath, which simultaneously targets several essential genes, as compared to siScrambled (*Figure 3—figure supplement 1a*), and reduction in NCAPH2 protein upon transfection of siNCAPH2 (*Figure 3—figure supplement 1b*)

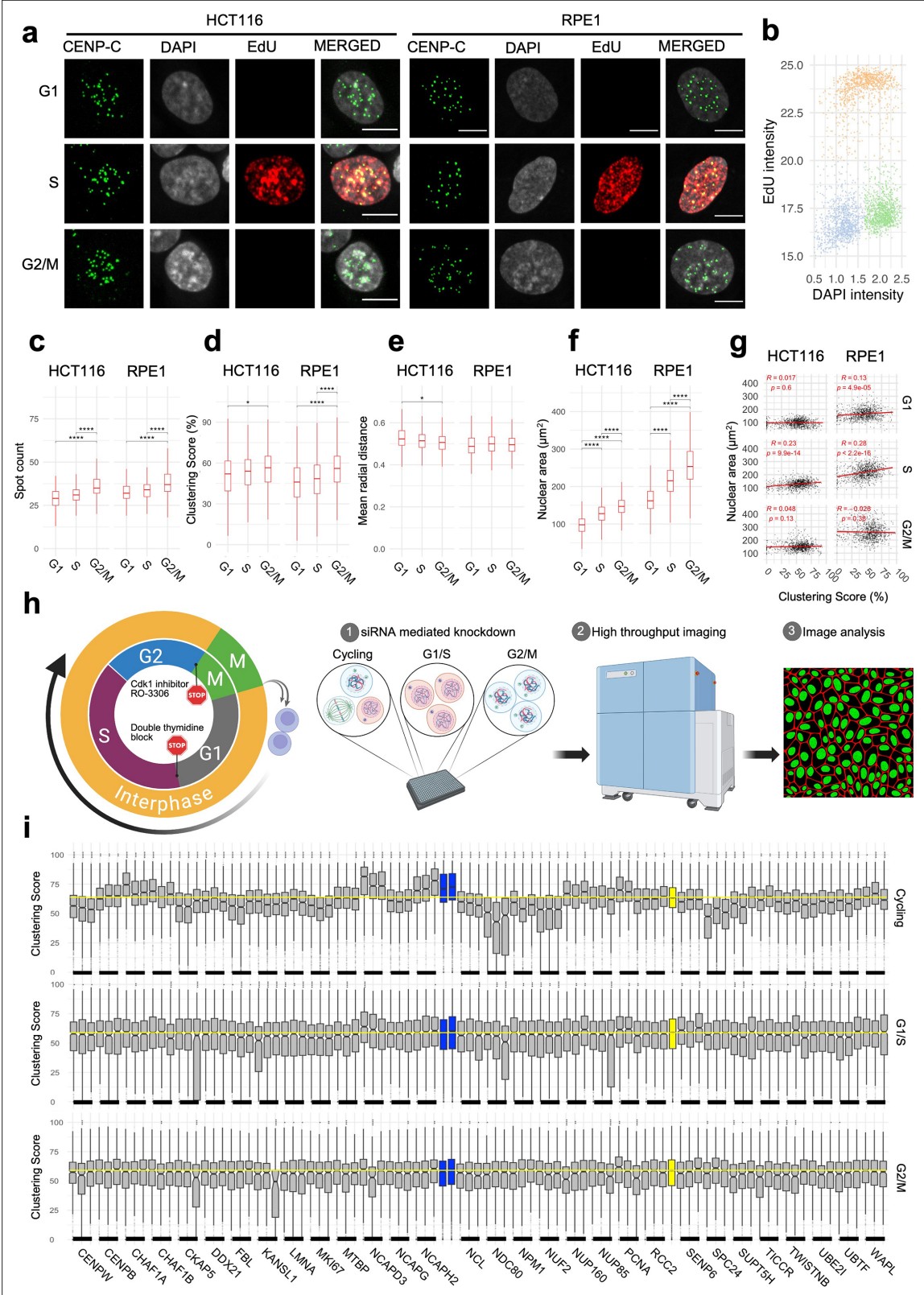

**Figure 3.** Changes in spatial organization of centromeres require progression through the cell cycle. (**a**) HCT116 and RPE1 cells in G1, S, or G2/M phases stained with CENP-C (green), DAPI (gray), and EdU (red). Scale bar: 10 µm. (**b**) EdU intensity (y-axis) and DAPI intensity (x-axis) showing separation between G1 (brown), S (gray), and G2/M (green) subpopulations in cycling HCT116 cells. Comparison of cells in G1, S, or G2/M (x-axis) for their clustering score (**c**), spot count (**d**), mean radial distance (**e**), or nuclear area (**f**). Statistical significance of differences was tested by pairwise t-test

*Figure 3 continued on next page*

*Figure 3 continued*

with Bonferroni correction. Asterisks indicate level of significance between a given pair reflecting the corresponding p-value of that comparison. (**g**) A linear regression line (red) fitted through the single-cell data for nuclear area (y-axis) and clustering score (x-axis) in cells in different cell-cycle phases in HCT116 and RPE1 cells. Pearson's correlation coefficient and respective adjusted p-values are indicated at the top of each panel. (**h**) Experimental outline to test cell-cycle stage-specific effect of knocking down select hits. (**i**) Effect of siRNA knockdown for a panel of genes (x-axis) using three individual siRNAs per gene in HCT116 cells that are either arrested at G/S and G2 or cycling. Two control siRNAs for siNCAPH2 are in blue and siScrambled in yellow. The mean value for siScrambled is depicted by a horizontal yellow dotted line. Statistical significance of differences was tested by performing pairwise t-tests with Bonferroni correction using siScrambled as control group. Box plots represent the interquartile range (IQR) between the first and third quartiles (box), the median (horizontal bar), and the whiskers that extend to the highest and lowest value within 1.5 times the IQR. Values are from one representative experiment. Typically, 200–500 cells were analyzed in each category. Statistical significance of difference was denoted by stars where * indicates p≤0.05, ** indicates p≤0.01, *** indicates p≤0.001, and **** indicates p<0.0001.

The online version of this article includes the following source data and figure supplement(s) for figure 3:

**Figure supplement 1.** Validation of protein depletion in siRNA knockdown.

**Figure supplement 1—source data 1.** Original western blot images used in *Figure 3—figure supplement 1*.

**Figure supplement 1—source data 2.** PDF file containing annotated western blot images used in *Figure 3—figure supplement 1*, with figure legend explaining relevant details.

indicates efficient siRNA knockdown in cycling, G1/S, and G2/M cells. While centromere distribution was altered upon knockdown of these genes in cycling cells as expected, no changes in centromere distribution were observed when knockdowns were done in G1/S or G2/M arrested cells (*Figure 3h*). We conclude that while the distribution of centromeres does not vary during the cell cycle, progression through the cell cycle is required to bring about changes in centromere distribution in the absence of key regulators of centromere clustering. These results demonstrate that the identified modifiers of centromere distribution do not act in the maintenance of centromere distribution during interphase.

## Normal progression through mitosis is required for faithful interphase centromere distribution

Having established that cell-cycle progression is required for the effects of the identified centromere distribution factors, we asked at what stage of the cell cycle the centromere distribution factors act. We measured changes in clustering score before and after progressing through either S-phase or mitosis in cells depleted of a given factor (*Figure 4a*). We selected four proteins for this analysis that all interact with centromeres and contribute to efficient and error-free chromosome segregation during mitosis but all have distinct functions: NCAPH2 is a component of the Condensin II complex and responsible for axial compaction of chromosomes (*Shintomi and Hirano, 2011*; *Green et al., 2012*; *Gibcus et al., 2018*); KI67 is an established marker of cell proliferation that decorates nucleoli in interphase cells and coats chromosomes during mitosis (*Booth et al., 2014*; *Cuylen et al., 2016*); SPC24 and NUF2 are kinetochore components and part of the NDC80 complex which connects the kinetochore to microtubules (*McCleland et al., 2004*; *Ciferri et al., 2008*). Auxin-inducible degron cell lines to deplete NCAPH2 or KI67 have previously been characterized (*Takagi et al., 2018*). In addition, we generated dTAG-SPC24 and NUF2-dTAG cell lines by CRISPR knock-in into HCT116-Cas9 parental cells (*Figure 4—figure supplement 1a and b*; Methods). Homozygous knock-in in select clones was verified by PCR genotyping (*Figure 4—figure supplement 1c and d*), and correct localization and expression of dTAG-SPC24 and NUF2-dTAG as compared to the parental cell line was confirmed by indirect IF staining of the tagged proteins and by western blotting, respectively (*Figure 4—figure supplement 2a, b, e, and f*; *McCleland et al., 2004*). Effective depletion of each factor by more than 90% as assessed by western blotting was achieved within 3 hrs (*Figure 4—figure supplement 2c and d*).

Using these four HCT116-based degron lines, we acutely depleted individual factors specifically in cells arrested at the G1/S or G2/M boundaries and then released the cell-cycle block (see Methods). We validated effective depletion of SPC24 and NUF2 in G1/S and G2/M arrested cells (*Figure 4—figure supplement 3a, b, c, and d*). First, we compared clustering scores in cells progressing through S-phase in the presence or absence of NCAPH2, KI67, SPC24, or NUF2 (*Figure 4a*). Upon release from a standard double thymidine block, the majority (58–78%) of cells reached G2/M after 6 hrs

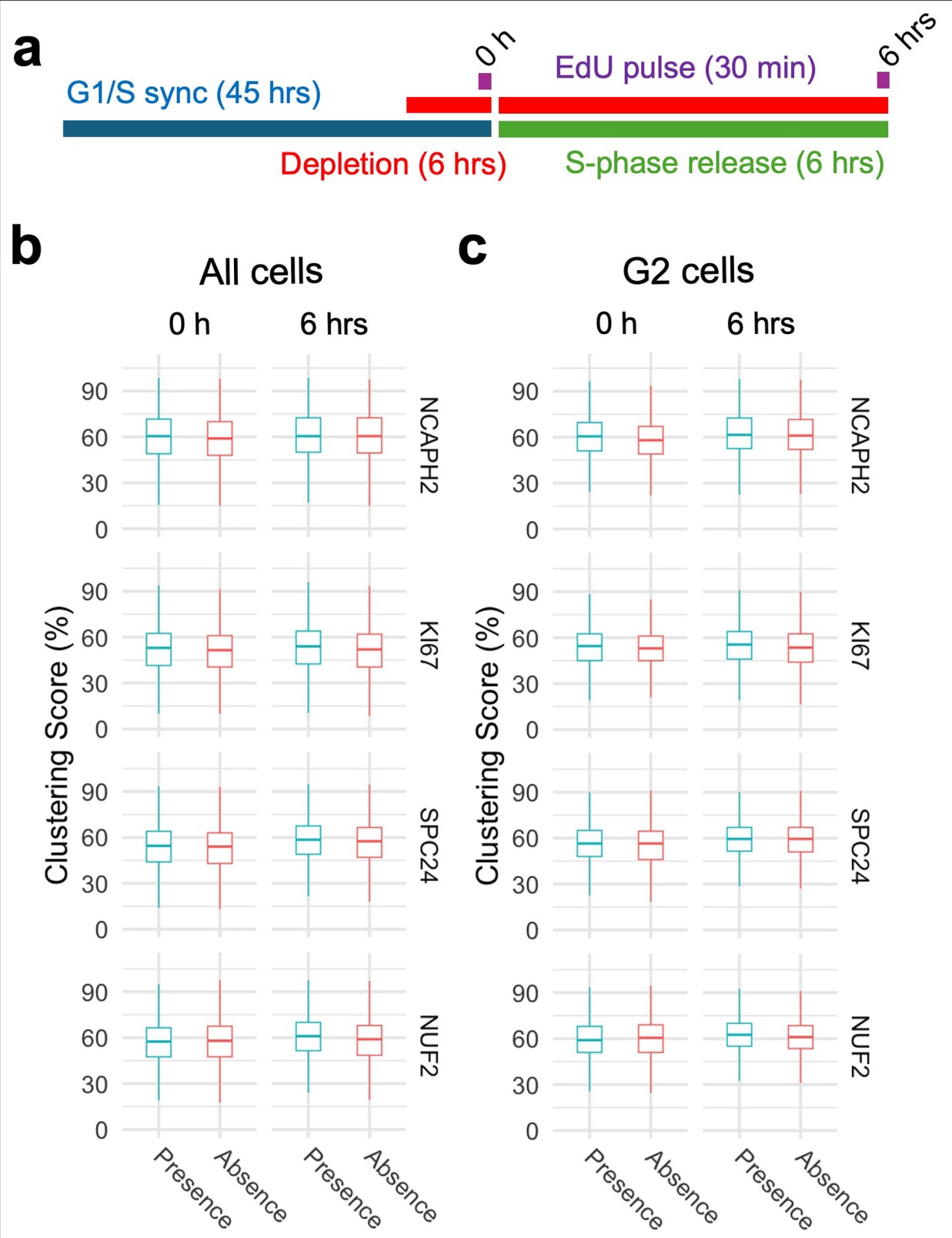

**Figure 4.** Progression through S-phase in the absence of select clustering factors does not alter interphase genome organization. (**a**) Experimental outline to compare centromere distribution during progression through S-phase in the presence or absence of clustering factors. (**b, c**) Clustering score in all cells (**b**) or G2/M cells (**c**) in the presence (blue) or absence (red) of indicated clustering factors before and after S-phase release from G1/S arrest. Pairwise comparisons were performed using t-tests with Bonferroni correction, and the level of significance is indicated by asterisks if any. Pairs without

*Figure 4 continued on next page*

*Figure 4 continued*

significant difference are not labeled. Box plots represent the interquartile range (IQR) between the first and third quartiles (box), the median (horizontal bar), and the whiskers that extend to the highest and lowest value within 1.5 times the IQR. Values are from one representative experiment with three technical replicates. Typically, 200–500 cells were analyzed in each category.

The online version of this article includes the following source data and figure supplement(s) for figure 4:

**Figure supplement 1.** Construction and genotyping of FLAG-dTAG-SPC24 and NUF2-dTAG-FLAG cell lines.

**Figure supplement 1—source data 1.** Original western blot images used in *Figure 4—figure supplement 1*.

**Figure supplement 1—source data 2.** PDF file containing annotated western blot images used in *Figure 4—figure supplement 1*, with figure legend explaining relevant details.

**Figure supplement 2.** Characterization of FLAG-dTAG-SPC24 and NUF2-dTAG-FLAG cell lines.

**Figure supplement 2—source data 1.** Original western blot images used in *Figure 4—figure supplement 2*.

**Figure supplement 2—source data 2.** PDF file containing annotated western blot images used in *Figure 4—figure supplement 2*, with figure legend explaining relevant details.

**Figure supplement 3.** Quantification of the cell-cycle stage-specific depletion of FLAG-dTAG-SPC24 and NUF2-dTAG-FLAG.

**Figure supplement 3—source data 1.** Original western blot images used in *Figure 4—figure supplement 3*.

**Figure supplement 3—source data 2.** PDF file containing annotated western blot images used in *Figure 4—figure supplement 3*, with figure legend explaining relevant details.

**Figure supplement 4.** Quantification of cell-cycle stages during G1 and mitotic release.

(*Figure 4—figure supplement 4a and c*). Progression through S-phase was equally efficient in the presence or absence of KI67, SPC24, or NUF2 (*Figure 4—figure supplement 4a and c*). Acute NCAPH2 depletion mildly delayed S-phase progression (*Figure 4—figure supplement 4a and c*), as reported earlier (*Rodemoyer et al., 2025*). Although loss of KI67 has been reported to delay replication of centromeres and pericentromeric loci (*van Schaik et al., 2022*; *Stamatiou et al., 2024*), no effect on bulk S-phase progression after KI67 loss was observed in our hands (*Figure 4—figure supplement 4a*). No effect on clustering scores was evident as cells progressed through S-phase into G2, regardless of the presence or absence of any of these proteins (p>0.05; *Figure 4b and c*). We conclude that these centromere distribution modifiers do not act in S-phase.

Next, we tested if the loss of function of these centromere distribution modifiers during mitosis altered centromere localization in the subsequent interphase cells (*Figure 5a*). HCT116 cells were arrested at the G2/M boundary by treatment for 20 hrs with the CDK1 inhibitor RO-3306 as previously described (*Vassilev et al., 2006*), and then released for 6 hrs in the absence of each mitotic factor. The newly formed G1 cells were analyzed for centromere distribution. As expected, mitotic progression in the absence of SPC24 or NUF2 was slowed upon release from the G2/M block (*McCleland et al., 2004*) with 28–35% of cells reaching G1 after 6 hrs compared to 58–60% in the presence of SPC24 or NUF2 (*Figure 4—figure supplement 4b and d*) and some aberrant nuclear phenotypes were evident in the absence of SPC24 or NUF2 (*Figure 5—figure supplement 1a, b, c, and d*). Mitotic progression in the absence of NCAPH2 and KI67 was similar to that of control cells (*Figure 4—figure supplement 4b and d*). While the centromere distribution phenotypes remained unaltered compared to cycling cells in the presence of these proteins, progression through a single mitosis in the absence of any of these proteins altered centromere distribution phenotypes in the subsequent G1 phase as assessed by quantitation using the Ripley's K clustering score (*Figure 5b and c*) and visual inspection (*Figure 5d*). Loss of NCAPH2 had the largest effect and resulted in increased clustering of centromeres in G1 cells (*Figure 5c*; p<10$^{-10}$). Similarly, progression through mitosis in the absence of KI67 reduced clustering in the newly forming G1 cells (p=1.27e-08), as did loss of SPC24 (p<10$^{-10}$) or NUF2 (p<10$^{-10}$) (*Figure 5c*). We conclude that the function of NCAPH2, KI67, SPC24, and NUF2 during mitosis determines centromere distribution patterns in the newly formed daughter nuclei. The fact that loss of proteins with distinct mitotic functions perturbs centromere organization in the subsequent G1 phase suggests that, rather than their specific mitotic functions, it is the orderly progression of cells through mitosis that is required to ensure the faithful maintenance of spatial centromere distribution.

Since all four factors act during mitosis but had different effects on centromere distribution, we explored co-depletion phenotypes to understand functional overlap between these factors, if any. We combined siRNA knockdown and degron-based depletion of NCAPH2, KI67, or SPC24 in

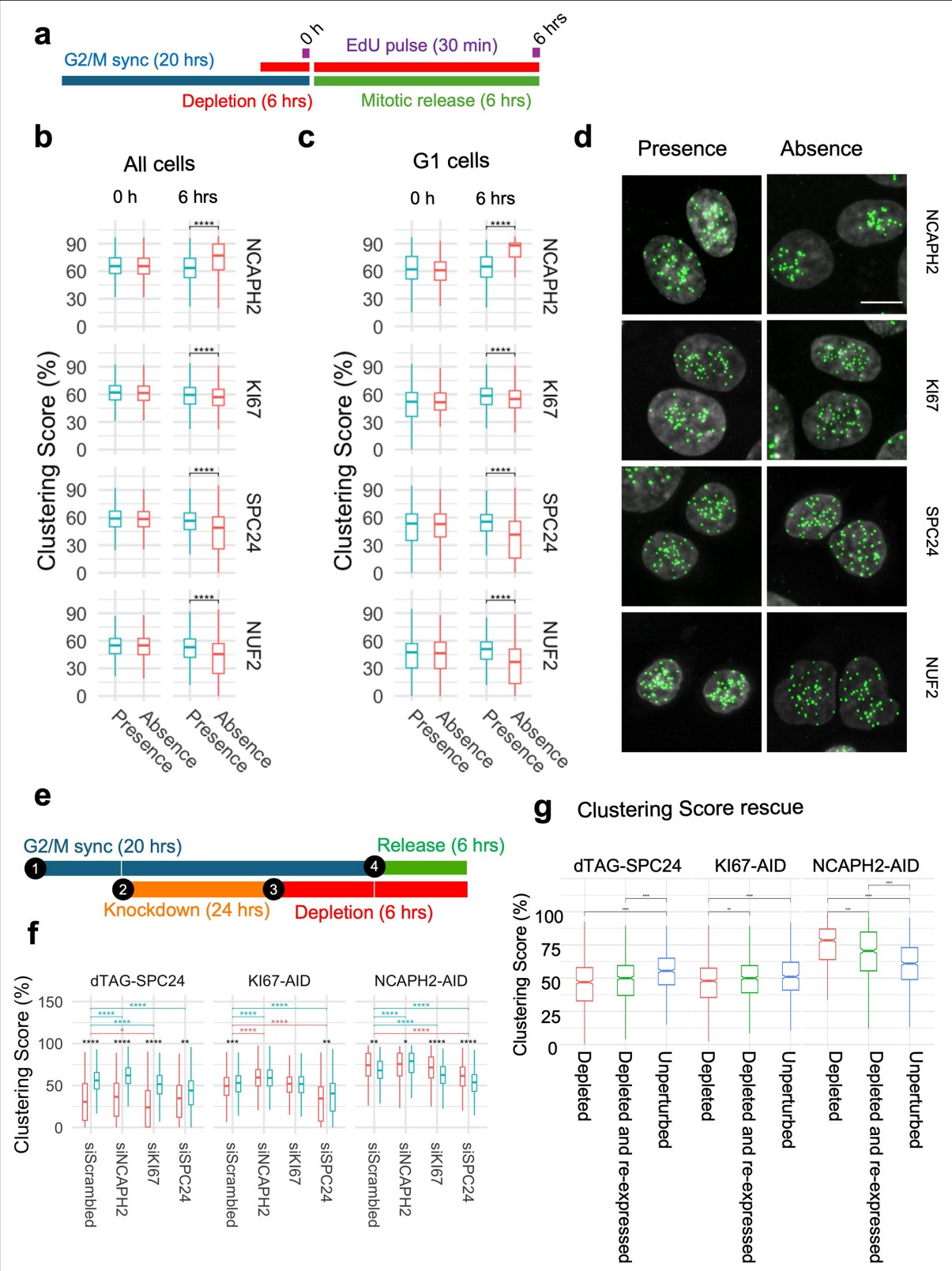

**Figure 5.** Orderly progression through mitosis is required for normal centromere distribution. (**a**) Experimental outline to compare centromere distribution during mitotic progression in the presence or absence of clustering factors. (**b, c**) Clustering score in all cells (**b**) or G1 cells (**c**) in the presence (blue) and absence (red) of indicated clustering factors before (0 hr) and after (6 hr) mitotic release from G2 arrest. Pairwise comparisons were performed using t-tests with Bonferroni correction, and the level of significance is indicated by asterisks. Pairs without significant differences are not

*Figure 5 continued on next page*

*Figure 5 continued*

labeled. (**d**) Representative images showing G1 nuclei stained with DAPI (gray) and CENP-C (green) in the presence or absence of indicated factors. Scale bar: 10 µm. (**e**) Schematics for co-depletion of indicated factors. (**f**) Clustering score (y-axis) in G1 cells after siRNA knockdown of indicated factors (x-axis) in presence (blue) or absence (red) of SPC24, KI67, or NCAPH2 as indicated. Statistical significance of difference between indicated pairs was tested by performing t-test with Bonferroni corrections for multiple comparisons and denoted by stars, where * indicates p≤0.05, ** indicates p≤0.01, *** indicates p≤0.001, and **** indicates p<0.0001. Comparisons between individual siRNA groups (x-axis) and inter-group comparisons in the presence and absence of the given degron tagged protein are shown in black, blue (presence), and red (absence), respectively. (**g**) Clustering score (y-axis) in cells that were depleted (red) or depleted of indicated factors and then re-expressed (green) or remained unperturbed (blue). Statistical significance of difference between indicated pairs was tested by performing t-test with Bonferroni corrections for multiple comparisons and denoted by stars, where a higher number of stars indicate higher confidence levels. Box plots represent the interquartile range (IQR) between the first and third quartiles (box), the median (horizontal bar), and the whiskers that extend to the highest and lowest value within 1.5 times the IQR. Values are from one representative experiment containing three technical replicates. Typically, 200–500 cells were analyzed for each category. Statistical significance of difference was denoted by stars where * indicates p≤0.05, ** indicates p≤0.01, *** indicates p≤0.001, and **** indicates p<0.0001.

The online version of this article includes the following figure supplement(s) for figure 5:

**Figure supplement 1.** Mitotic defects in the absence of SPC24.

---

pairwise combinations along with scrambled siRNAs and non-depleted cells as controls (*Figure 5e*). We observed additive effects as simultaneous loss of SPC24 and KI67 further reduced the clustering score than the individual loss of either KI67 (p=5.0640e-47) or SPC24 (p=5.8800e-03). Similarly, the centromere unclustering upon KI67 knockdown was rescued by simultaneous NCAPH2 depletion (*Figure 5f*; p=3.2280e-10). In contrast, centromeres did not cluster more when *NCAPH2* was either knocked down (p=5.9760e-56) or depleted (p=3.2280e-10) in the absence of *SPC24* (*Figure 5f*), indicating that SPC24 functions upstream of NCAPH2 in regulating spatial centromere position. These findings point to an intricate interplay of these factors and pathways in determining centromere positioning.

We finally asked whether the aberrant altered centromere distribution in daughter cells upon depletion of mitotic factors can be reversed upon re-expression of NCAPH2, KI67, or SPC24. To test this idea, each of these factors was depleted for 6 hrs in asynchronous cells following washout of degron ligands to allow re-expression of NCAPH2, KI67, or SPC24 as they progress through the cell cycle for 24 hr. Cells with and without depletion are used as controls. We observed partial rescue upon re-expression of all three factors as clustering scores partially returned toward that of the unperturbed cells (*Figure 5g*).

Taken together, these findings demonstrate a requirement for orderly progression through mitosis for the faithful establishment of the spatial distribution of centromeres and global genome organization in the subsequent interphase nuclei.

## Discussion

We identify here several cellular factors that determine the 3D positions of centromeres in the human cell nucleus, and we find that interference with orderly progression through mitosis alters centromere location in the subsequent interphase. We conclude that mitotic events shape the spatial organization of the interphase genome.

The most prominent group of centromere distribution effectors were components of the mitotic machinery, particularly multiple kinetochore proteins, including all four components of the NDC80 complex (*McCleland et al., 2004*) and components of the CENP-T-W-S-X complex (*Nishino et al., 2012*). The fact that loss of multiple factors with distinct mechanisms of action, but all affecting mitosis, resulted in altered centromere distribution in the newly formed G1 cells points to a prominent role for orderly progression through mitosis as a main determinant of interphase centromere distribution, reminiscent of prior observations on lamina-associated chromatin domains which stochastically reposition during mitosis (*Kind et al., 2013*).

A likely mechanism for the observed altered arrangement of centromeres in early G1 upon interference with mitotic machinery is the aberrant alignment of chromosomes in the mitotic plate and their uncoordinated migration toward the spindle poles (*Figure 6*). As cells enter mitosis, the outer kinetochores assemble on the centromeres, chromosomes condense and align on the metaphase plate. This process is initiated in late G2 when the KMN (KNL1, MIS12, and NDC80) complex, including the

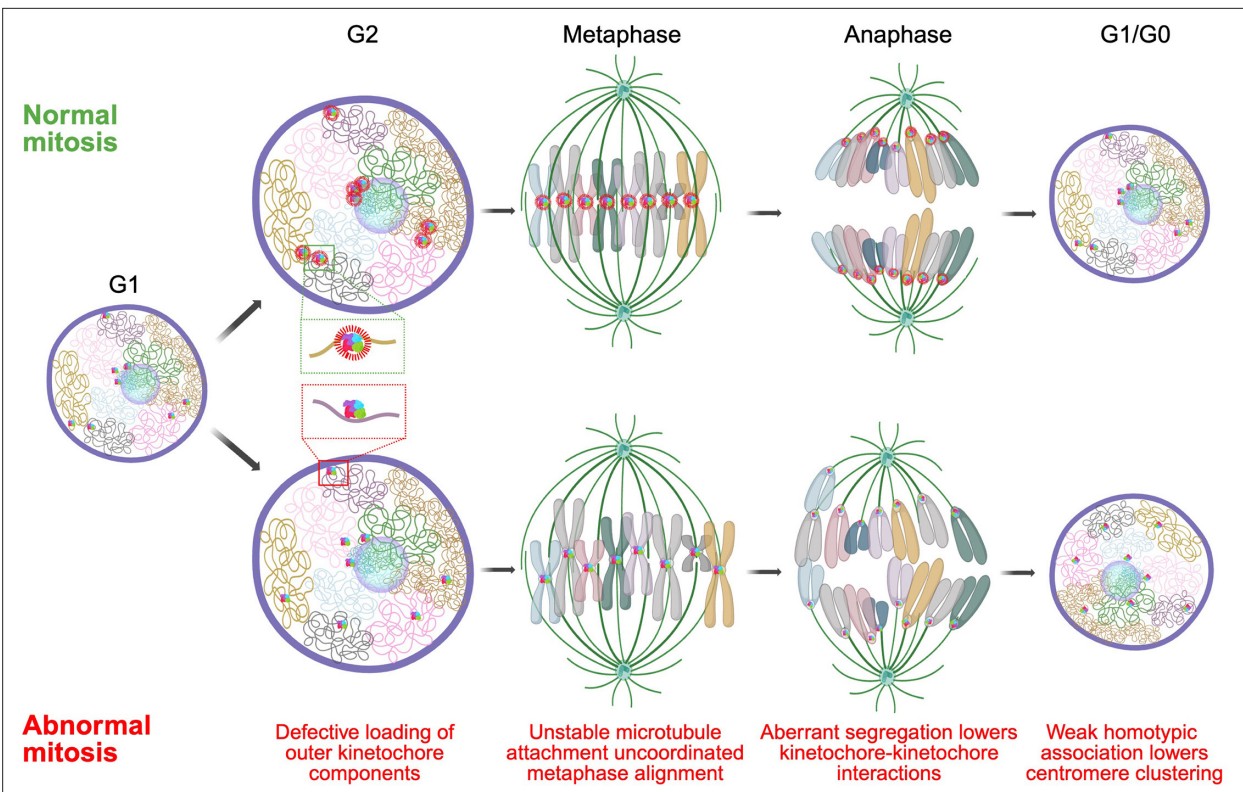

**Figure 6.** Mitotic events shape interphase genome organization. A model showing defective loading of outer kinetochore (inset; green and red box) in late G2 leads to uncoordinated metaphase alignment and aberrant migration toward spindle pole during anaphase that lowers chances of interactions between centromeres during telophase, and the lack of homotypic adhesion results in dispersion of centromeres in the daughter nuclei.

NDC80 complex, is loaded onto the kinetochore to stabilize microtubule attachments (*Gascoigne and Cheeseman, 2013*). Loss of NDC80 components, such as SPC24 or NUF2, weakens microtubule attachments but does not completely disrupt chromosome segregation as has been observed for CENP-A, CENP-C, and other components of inner kinetochore (*Ciferri et al., 2007*). As such, chromosomes will progress through mitosis but will be imprecisely oriented in the metaphase plate and will migrate in an uncoordinated fashion to the spindle poles, leading to their dispersal in early G1. Indeed, we find that the assembly and disassembly of the NDC80 complex correlates with lower clustering score in G1 cells compared to G2/M cells in a cycling population. This effect is further exaggerated upon KO or depletion of multiple NDC80 complex components resulting in stronger centromere dispersion. The observed mitotic effects on interphase organization are reminiscent of recent observations on the relationship of chromosome location and mis-segregation defects (*Klaasen et al., 2022*) where single-cell observations indicated that the more peripheral a chromosome is in the interphase nucleus, the higher its chance of improper alignment in the metaphase plate and consequently being mis-segregated leading to aneuploidy (*Klaasen et al., 2022*; *Vukušić and Tolić, 2022*). Similarly, the observed effect of NCAPH2 depletion on centromere distribution may reflect a defect in chromosome segregation. Loss of NCAPH2 has been shown to lengthen chromosomes which may facilitate homotypic centromere-centromere interactions, resulting in the observed increase in clustering of centromeres in G1 (*Hoencamp et al., 2021*). A further contributor to the mitotic effect on interphase centromere distribution may be defects in mitotic exit, as suggested by our identification of KI67 as a determinant of centromere distribution. KI67 has been localized to centromeres (*van Schaik et al., 2022*) and reported to act in late telophase as a surfactant to generate mechanical forces that are required for re-establishing nuclear-cytoplasmic compartmentalization in G1 cells (*Cuylen et al., 2016*; *Hernandez-Armendariz et al., 2024*). Loss of KI67 may disrupt the arrangement and progression of chromosomes in late telophase, leading to redistribution of centromeres in G1. Pairwise depletion of

these factors producing additive effects on clustering pointed to an intricate interplay of the affected pathways in determining interphase centromere positioning.

Regardless of the precise molecular mechanisms for how mitotic progression affects the distribution of centromeres in interphase, it is likely that the location of centromeres in the nuclear space is in large parts driven by homotypic interactions. Centromeres are specialized genomic loci that are highly heterochromatic with low transcription activity (*Altemose et al., 2022*). It is well established that homotypic chromatin regions, such as heterochromatin, self-interact as is evident by the formation of the A and B chromatin compartments, which incorporate regions of similar chromatin status from distinct chromosomes (*Hildebrand and Dekker, 2020*; *Misteli, 2020*). A homotypic self-organization model for centromeres is in line with the presence of chromocenters in mouse, fly, and plant cells, which represent clusters of peri-centromeric regions from multiple chromosomes forming large heterochromatin blocks (*Sullivan and Karpen, 2004*; *Probst and Almouzni, 2011*; *Jagannathan et al., 2019*), but are largely absent in humans, indicating factors implicated in chromocenter maintenance are probably not required for spatial centromere organization in humans. Indeed, we find that loss of *HMGA1*, whose gene product stabilizes chromocenters in mouse (*Jagannathan et al., 2018*), or other HMG genes, did not affect the spatial distribution of centromeres in human cells, suggesting species specificity of some determinants of genome organization. A heterochromatin-driven homotypic interaction model also explains the prominent association of centromeres with the nucleolus in human stem cells, which are largely devoid of nuclear heterochromatin blocks (*Meshorer and Misteli, 2006*), making the nucleolus the most prominent high-affinity binding site for centromeres in the nucleus. Our model is also in line with the long-standing observation that following mitosis, chromosome unfolding leads to the re-establishment of the chromatin landscape of interphase nuclei (*Belmont and Bruce, 1994*).

It is intriguing to speculate that altered centromere position may have functional consequences. For example, dispersion of centromeres may increase centromere to non-centromere contacts, thus influencing the local chromatin environment of both repositioned centromere and non-centromere loci, possibly altering their transcriptional output. On the other hand, clustering of centromeres may help preserve integrity of the centromeric and peri-centromeric chromatin environment but may also increase the likelihood of inter-centromere translocations, thereby disrupting genome stability. Finally, identification of centromere position regulators now allows experimental perturbation of higher-order genome organization and tests its impact on overall gene expression and genomic stability, adding to our current understanding of the broad connection between genome structure and function.

## Methods

**Key resources table**

| Reagent type (species) or resource | Designation | Source or reference | Identifiers | Additional information |
|---|---|---|---|---|
| Cell line (*Homo sapiens*) | Cell line FLAG-dTAG-SPC24 S8 | This study | | Parental cell line HCT116 Cas9 |
| Cell line (*Homo sapiens*) | Cell line NUF2-dTAG-FLAG N10 | This study | | Parental cell line HCT116 Cas9 |
| Cell line (*Homo sapiens*) | Cell line NCAPH2-mACh | *Takagi et al., 2018* | | Parental cell line HCT116 |
| Cell line (*Homo sapiens*) | Cell line KI-67-mACl | *Takagi et al., 2018* | | Parental cell line HCT116 |
| Cell line (*Homo sapiens*) | Cell line HCT116 Cas9 | *Hart et al., 2015* | | Human colon cancer |
| Cell line (*Homo sapiens*) | Cell line A375 Cas9 | *Hart et al., 2015* | | Near triploid, Human malignant melanoma |
| Cell line (*Homo sapiens*) | Cell line hTERT-RPE1 Cas9 | *Hart et al., 2015* | | Immortalized human Retinal pigment epithelia |
| Cell line (*Homo sapiens*) | Cell line WTC-11 | Coriell Institute | GM25256 | Human induced pluripotent stem cell (iPSC) |

*Continued on next page*

*Continued*

| Reagent type (species) or resource | Designation | Source or reference | Identifiers | Additional information |
|---|---|---|---|---|
| Cell line (*Homo sapiens*) | Cell line MDA-MB-231 Cas9 | Horizon Discovery | HD Cas9-014 | Human triple-negative breast cancer |
| Cell line (*Homo sapiens*) | Cell line HAP1 Cas9 | Horizon Discovery | HD Cas9-011 | Near haploid, chronic myelogenous leukemia |
| Cell line (*Homo sapiens*) | Cell line A549 Cas9 | Horizon Discovery | HD Cas9-001 | Human lung adenocarcinoma |
| Cell line (*Homo sapiens*) | Cell line HFF-hTERT | Clone 6, Dekker lab; 4DNucleome project cell line | | Immortalized human foreskin fibroblast |
| Recombinant DNA reagent | Plasmid pJT142 | This study | Available from Addgene | Cas9 and sgRNA targeting SPC24 |
| Recombinant DNA reagent | Plasmid pMG1040 | This study | Available from Addgene | Cas9 and sgRNA targeting NUF2 |
| Recombinant DNA reagent | Plasmid pJT152 | This study | Available from Addgene | Donor for FLAG-dTAG-SPC24 |
| Recombinant DNA reagent | Plasmid pMG1064 | This study | Available from Addgene | Donor for NUF2-dTAG-FLAG |
| Antibody | Guinea pig polyclonal anti-CENP-C | MBL Biosciences | PD030 | 1:1000 dilution for IF staining |
| Antibody | Mouse monoclonal anti-CENP-A | Abcam | AB13939 | 1:1000 dilution for IF staining |
| Antibody | Mouse monoclonal anti-FLAG | Sigma | F3165 | 1:250 dilution for IF staining |
| Antibody | Mouse monoclonal anti-β-Actin | Millipore Sigma | A2228 | 1:25,000 dilution for western blotting |
| Antibody | Rabbit polyclonal anti-SPC24 | *McCleland et al., 2004* | Gift from Todd Stukenberg lab | 1:8000 dilution for western blotting |
| Antibody | Rabbit monoclonal anti-NUF2 | Abcam | ab176556 | 1:2000 dilution for western blotting |
| Chemical compound | RO-3306 | Millipore Sigma | Cat. No. 217721 | Working concentration 9 µM |
| Chemical compound | Auxin | Millipore Sigma | I3750-25G-A | Working concentration 500 nM |
| Chemical compound | dTAG13 | Tocris Bioscience | Cat. No. 6605 | Working concentration 1 µM |
| Chemical compound | dTAG$^V$-1 | Tocris Bioscience | Cat. No. 6914 | Working concentration 1 µM |
| Chemical compound | Thymidine | Sigma | Cat. No. T9250-5G | Working concentration 2 mM |
| Chemical compound | Lipofectamine | Thermo Fisher Scientific | Cat. No. 13778075 | Transfection reagent |

## Cell culture

All cell lines used in this study are grown in a humidified 37°C incubator in the presence of 5% $CO_2$. Sources, composition of growth media, relevant references of the respective cell culture protocols are provided in *Supplementary file 1*. Cells were fixed with 2% paraformaldehyde (PFA, Electron Microscopy Sciences, Cat. No. 15710) solution in media by adding one equal volume of 4% PFA solution in PBS to the cell growth medium in each well for 15 min at room temperature. To activate protein degradation, 500 nM auxin (Millipore Sigma, Cat. No. I3750-25G-A) or 1 µM dTAG13 (Tocris Bioscience, Cat. No. 6605) or 1 µM dTAG$^V$-1 (Tocris Bioscience, Cat. No. 6914) ligand was used. The identity of all cell lines has been authenticated by sequencing and tested periodically for absence of mycoplasma. Details on cell lines are provided in the Methods section. Cell lines: FLAG-dTAG-SPC24 (this study); NUF2-dTAG-FLAG (this study); NCAPH2-mACh *Takagi et al., 2018*; KI-67-mACl *Takagi et al., 2018*; HCT116 Cas9 *Hart et al., 2015*; A375 Cas9 *Hart et al., 2015*; hTERT-RPE1 Cas9 *Hart*

*et al., 2015*; WTC-11 (Coriell Institute, GM25256); MDA-MB-231 Cas9 (Horizon Discovery, HD Cas9-014); HAP1 Cas9 (Horizon Discovery, HD Cas9-011); A549 Cas9 (Horizon Discovery, HD Cas9-001); and HFF-hTERT (Clone 6 4DNucleome project cell line).

## Cell-cycle synchronization

Cells were synchronized at the G1/S boundary by double thymidine block as described (*Chen and Deng, 2018*). Briefly, cells were seeded at 20–30% confluency and grown for 24 hrs following 18 hrs of growth in media containing 2 mM thymidine (Sigma, Cat. No. T9250-5G). Cells were then washed with PBS and grown in fresh media for 9 hrs. The growth media was changed with fresh media containing 2 mM thymidine. Cells were synchronized at the G2/M stage by treating cells with 9 µM RO-3306 (Millipore Sigma, Cat No. 217721-2MG) for 20 hrs as described (*Vassilev et al., 2006*). Cells were once washed in 300 µL fresh prewarmed growth media and replenished with fresh growth media to release from the G1/S or G2/M block.

## IF staining

Indirect IF staining was performed as previously described (*Keikhosravi et al., 2024*; *Keikhosravi et al., 2025b*). Briefly, fixed cells grown on a 96-well plate (Revvity, Cat. No. 6055300) or 384-well plate (CellVis, Cat. No. P384-1.5H-N) were washed with PBS and then permeabilized with 0.1% Triton X-100 (Sigma, Cat. No. T9284-500ML) solution in PBS for 15 min and again washed with PBS. These cells were blocked by incubating with 5% BSA (Millipore Sigma, Cat. No. A3294-100G) solution in PBST (0.05% Tween-20 in PBS) for 15 min at room temperature. Next, cells were incubated with appropriate primary antibody dilution prepared in blocking solution (5% BSA in PBST) for 1 hr at room temperature and then washed with PBS three times. Next, the cells were incubated for 1 hr with fluorescently labeled secondary antibody solution prepared in blocking solution at room temperature and then washed with PBS three times. DAPI (4',6-diamidino-2-phenylindole) (Thermo Fisher Scientific, Cat. No. 62248) staining was performed by adding 5 µg/mL DAPI solution prepared in 1× PBS to the wells. Anti-CENP-C (MBL Biosciences, Cat. No. PD030) and anti-CENP-A (Abcam, Cat. No. AB13939) antibodies were diluted 1:1000 in blocking buffer (5% BSA solution prepared in PBST) and used for IF staining. In experiments where cells were EdU labeled, CENP-C primary antibody was directly conjugated with fluorophores using Mix-n-Stain CF Dye Antibody Labeling Kit (Biotium, Cat. No. 922235). Anti-FLAG monoclonal antibody (Sigma, Cat. No. F3165250) was diluted 1:250 in blocking buffer and used for IF staining.

## CRISPR-KO library design

A custom arrayed synthetic sgRNA library targeting 1064 genes associated with chromatin biology and nuclear architecture was sourced from Synthego (Cat # SO17105 and 8311960), delivered lyophilized in 96-well plates, resuspended in RNAse-free ddH$_2$O, and reformatted in 384-well plate format using the PerkinElmer Janus and the Beckman Coulter ECHO525 liquid handlers at a final concentration of 0.25 pmoles/µL. Each gene was targeted in the same well by three pooled sgRNA oligos that included the Synthego-modified EZ Scaffold. The list of genes and sgRNA targeting sequences in the library is included in *Supplementary file 3*.

## CRISPR-KO screens

For reverse transfection of sgRNA oligos in 384-well format, 325 nL of library sgRNA 0.25 pmoles/µL were spotted in each empty well (0.08 pmoles/well) of an imaging plate (CellVis, Cat. No. P384-1.5H-N) using an ECHO525 acoustic liquid handler. As controls, and in each plate, we also spotted 7 wells each of non-targeting scrambled control sgRNA (Synthego, Cat. No. 063-1010-000-000), sgRNAs targeting each of *PLK1, OR10A5*, and *NCAPH2*. Three sgRNAs pooled together in gene KO kits to target *PLK1*, *OR10A5,* and *NCAPH2* were obtained from Synthego (Cat. No. GKO-HS1-000-0-1.5n-0–0). The control sgRNAs had the same chemistry and were spotted in the same quantities as the sgRNAs in the library. Spotted plates were dried at room temperature under a laminar flow cell culture hood, sealed, and stored at –30°C until the day of the reverse transfection.

The day of the transfection, the spotted sgRNA imaging plates were thawed and equilibrated at room temperature and then spun at 1400 rpm. The seal was removed, and 20 µL of prewarmed serum-free Opti-MEM media (Thermo Fisher Scientific, Cat. No. 31985070) was dispensed into each well for

the imaging plate using a Thermo Fisher Multidrop dispenser. The ECHO525 was then used to dispense the required amount of Lipofectamine RNAi MAX (Thermo Fisher Scientific, Cat. No. 13778075). The plates were then incubated at room temperature for 30 min to allow RNA-Lipofectamine complexes to form. Next, a cell suspension prepared in prewarmed Opti-MEM media (Thermo Fisher Scientific, Cat. No. 31985070) containing 20% FBS was dispensed into each well using the Multidrop for a total volume of 40 μL and an effective final sgRNA concentration of 2 nM. Plates were incubated at room temperature inside a laminar airflow hood for 30 min before they were transferred into cell culture incubator and allowed to grow for 72 hr. CRISPR-KO screens were performed each in 2 biological replicates on different days.

## Imaging

384-Well plates containing fixed cells were then stained for CENP-C and DAPI and imaged using a Yokogawa CV8000 spinning disk confocal microscope. Imaging parameters were as described before (*Keikhosravi et al., 2024*). Briefly, IF images were collected on a multi-laser platform equipped with 405 nm (DAPI), 488 nm (for green fluorophores), 561 nm (for red fluorophores), and 640 nm (for far-red fluorophores) excitation lines that were combined through a 405/488/561/640 nm quad-band dichroic. Fluorescence was captured through a 60× water-immersion objective (NA = 1.2) and routed to either a 445/45 nm band-pass filter for DAPI or a 525/50 nm band-pass filter for green fluorophores, or 600/37 nm band-pass filter for red fluorophores or 676/29 nm band-pass filter for far-red fluorophores. A 16-bit sCMOS detector (2048×2048 pixels, 1×1 binning; effective pixel size = 0.108 μm) recorded Z-stacks with 1 μm steps, while real-time maximal projection was applied. A variable number of fields ranging from 9 to 22 were imaged per well in different imaging experiments to acquire sufficient number of cell images.

## Image analysis

Image analysis was performed as described (*Keikhosravi et al., 2025a*). Raw image stacks were processed using HiTIPS, our previously described high-content analysis pipeline for fixed- and live-cell assay (*Keikhosravi et al., 2024*). Max-projected DAPI channels provided nuclear masks, whereas CENP-A or CENP-C projections served for centromere spot localization (see *Figure 1—figure supplement 1B*). Analysis settings in HiTIPS were adjusted to the typical nuclear diameter and the intensity/size characteristics of centromere foci. Nuclear segmentation employed the GPU-accelerated CellPose algorithm (*Pachitariu and Stringer, 2022*), and centromere detection used a Laplacian-of-Gaussian approach. Final spot coordinates were defined as the centroid of each segmented focus. Average nuclear fluorescence intensity for the DAPI (405 nm) and EdU (640 nm) channels was measured at the single-cell level. Clustering scores were calculated using a metric derived from Ripley's K function as described (*Keikhosravi et al., 2025b*).

## Identification of screen hits

Single-cell data obtained from HiTIPS (nucleus area, number of CENP-C spots per cell, Ripley's K clustering score, mean normalized CENP-C spot radial distance) were averaged on a per-well level. Per-well average values and nuclei counts were then used as an input for the screen statistical analysis using R 4.3.3 (*R Development Core Team, 2024*) and the cellHTS2 package (*Boutros et al., 2006*). Briefly, per-well raw measurements were normalized on a per-plate basis using the median value of the sgRNA library treatments and the Z-score method. All the per-plate normalized values in different library plates originated from a single biological replicate were further standardized using a robust version of the Z-score. The Z-score values for the same well and plate combination in different biological replicates were then averaged to obtain a mean Z-score.

Screen hits were identified as genes whose KO resulted in a mean Z-score either higher than 2.5 or lower than –2.5 for either number of CENP-C spots per cell or Ripley's K clustering score. Nuclei that were either abnormally shaped (solidity<0.85) or were abnormally small (area<30 μm$^2$) indicative of micronuclei were not used for analysis. We also excluded sgRNAs that resulted in high cytotoxicity (cell number Z-score<–2.5) or whose mean Z-score was smaller than the Z-score standard deviation of two replicates.

## DAPI and EdU labeling for cell-cycle profiling

EdU labeling was performed using a kit (Thermo Fisher Scientific, Cat. No. C10340) as per the manufacturer's instructions. Briefly, cells were incubated with 1 μg/mL EdU for 45 min before fixation.

Fixed cells were permeabilized and blocked as described for IF staining protocol. Next, genome-incorporated EdU molecules during replication were fluorescently labeled by performing a click chemistry reaction for 30 min at room temperature. Subsequently, cells were washed twice with 1% BSA solution in PBST and stained with 5 µg/mL of DAPI solution in PBS at room temperature for 1 hr.

## Image analysis for cell-cycle profiling

Total nuclear fluorescence of the DAPI and EdU channel was quantified using HiTIPS (*Keikhosravi et al., 2024*). The quantitation data was processed downstream using R packages and log2-transformed DAPI integrated intensity was used to distinguish the G1 and G2/M subpopulations. Similarly, EdU-integrated fluorescence intensity was log2-transformed, and cells with detectable EdU intensity were classified as S-phase cells. Cells with DAPI intensity either higher than G2 cells (>4N population) or lower than G1 cells (subG1 population) were not analyzed.

## Construction of dTAG-SPC24 and NUF2-dTAG cell lines

Candidate guide RNAs targeting the N-terminus of SPC24 and C-terminus of NUF2 were designed using *sgRNA Scorer 2.0* (*Chari et al., 2017*) and CRISPRor (*Concordet and Haeussler, 2018*; *Supplementary file 6*). Briefly, candidate guide RNAs were in vitro transcribed and tested for cutting activity in cells, using an approach previously described (*Gooden et al., 2021*). Based on both the highest indel frequency, as well as proximity to the point of insertion, candidate 207 was selected for SPC24, and candidates 286 and 289 were selected for NUF2. Oligonucleotides corresponding to these three guide RNAs were phosphorylated, annealed, and ligated into either the pX458 backbone (Addgene #48138) (*Ran et al., 2013*) for SPC24 (pJT142) or pDG458 (Addgene #100900) (*Adikusuma et al., 2017*) for NUF2 (pMG1040). pSpCas9(BB)-2A-GFP (PX458) was a gift from Feng Zhang (Addgene #48138). Plasmid pDG458 was a gift from Paul Thomas (Addgene #100900).

To generate the homology-directed repair donors for construction of dTAG-SPC24 and NUF2-dTAG cell lines, first, DNA fragments with 5′ and 3′ homology arms were synthesized using Twist Biosciences and subsequently cloned into the pGMC00018 using isothermal assembly (*Gibson et al., 2009*) to generate intermediate constructs pJT147 (SPC24) and pMG1063 (NUF2). Subsequently, the Puromycin-2A-dTAG-3X-FLAG and 3X-FLAG-dTAG-2A-Puromycin cassettes were then PCR amplified from an existing plasmid (contains 3X-FLAG tagged version of dTAG, generated from Addgene #91796 or #91793) and then cloned into pJT147 to generate pJT152 and pMG1063 to generate pMG1064, respectively, using isothermal assembly approach. The oligonucleotides used to generate these constructs are listed (*Supplementary file 6*). All plasmids were sequenced completely using nanopore sequencing.

To generate the dTAG-SPC24 homozygous knock-in line, HCT116 Cas9 cells were co-transfected with two constructs: plasmid pJT142 encoding sgRNAs targeting the SPC24 locus and Cas9, and pJT152 encoding a donor template to introduce dTAG-FLAG and puromycin resistance using Lipofectamine LTX with Plus reagent (Thermo Fisher Scientific, Cat. No. 15338100) as instructed by the manufacturer. Similarly, to generate NUF2-dTAG homozygous knock-in line, HCT116 cells were co-transfected with pMG1040 encoding sgRNAs targeting NUF2 locus and Cas9, and pMG1064 encoding a donor template to introduce FLAG-dTAG and puromycin resistance. 24 hrs after transfection, cells were washed with fresh media and grown in fresh media for 24 hr. Next, cells were selected for puromycin resistance by growing them in the presence of 1.5 µg/mL puromycin (Thermo Fisher Scientific, Cat. No. A1113803) for 72 hr. Every 24 hr, old media was replaced with fresh media containing puromycin. Cells were further expanded in the presence of puromycin. Depending on the efficiency of transfection, cells took 5–7 days to achieve confluence. The cells were then harvested by using trypsin (Gibco, Cat. No. 15050065). A portion of the cells was frozen to generate a polyclonal stock, and the remaining cells were taken forward for single-cell cloning.

## Single-cell cloning

To generate single-cell clones, 500 cells were plated on a 15 cm dish and allowed to form colonies in the presence of puromycin. Single colonies appeared in 7–10 days. To isolate single colonies, 20–30 single colonies were selected, and a cloning cylinder was placed around them. The bottom of the cloning cylinder was sealed with grease. For clone isolation, 20 µL trypsin was added to each cylinder

and cells from individual colonies were resuspended in separate wells in a 96-well plate to recover single-cell clones.

## PCR confirmation

Genomic DNA from single-cell clones was isolated using genomic DNA purification kit (Thermo Fisher Scientific, Cat. No. K0512) as recommended. To identify homozygously tagged dTAG-SPC24 clones, KG49 (5'AGCTCAGACTTACAGGCGTG3') and KG76 (5'TGATGGTGCTGATGGTTGCA3') primers were used to amplify the genomic region flanking the site of integration at the SPC24 locus. This primer pair is expected to amplify a 2925 bp fragment from the tagged allele and an 1821 bp fragment from the untagged allele (*Figure 4—figure supplement 1a and c*). Similarly, homozygously tagged NUF2-dTAG clones were identified by PCR analysis using primer pair KG37 (5'CTGCTTTTCTTCCCCCACTG3') and KG64 (5'AGAGGCAGCCTTTTCTCTGA3'). NUF2-dTAG alleles produced a 3046 bp amplicon while the untagged allele produced an amplicon of 1943 bp length (*Figure 4—figure supplement 1b and d*).

## Western blotting

Two PCR-confirmed single-cell clones were used for western blot analysis to validate depletion of FLAG-dTAG-SPC24 or NUF2-dTAG-FLAG. Cells were grown in the presence of 1 µM dTAG13 ligand for 3, 6, and 9 hrs. Cells grown in the absence of the ligand were grown as a control. Protein samples were prepared using Bio-Rad lysis buffer (Bio-Rad, Cat. No. 1610747) as per the manufacturer's instructions. Briefly, $10^6$ cells were collected, washed with PBS, and lysed in 100 µL lysis buffer (Bio-Rad, Cat. No. 1610747) to isolate proteins.

Anti-FLAG monoclonal antibody (Sigma, Cat. No. F3165) was diluted 1:10,000 in blocking buffer (5% BSA solution prepared in TBST) was used for detection of FLAG-dTAG-SPC24 and NUF2-dTAG-FLAG in western blots. For loading control, anti-beta-actin antibody (Millipore Sigma, Cat. No. A2228) was diluted 1:25,000 in blocking buffer and used in western blots. For comparison between untagged SPC24 and FLAG-dTAG-SPC24 protein levels, antibody against SPC24 (gift from Todd Stukenberg lab) (*McCleland et al., 2004*) was used at 1:8000 dilution in blocking buffer. For comparison between untagged NUF2 and NUF2-dTAG-FLAG protein levels, antibody against NUF2 (Abcam, Cat. No. ab176556) was used at 1:2000 dilution in blocking buffer.

## siRNA knockdown assay

siRNA knockdown of a select panel of genes identified in the screen was carried out by reverse transfecting siRNAs (*Supplementary file 7*, list of siRNAs used). Briefly, 150 nL of 5 µM siRNA stock solution was spotted on the 384-well glass-bottom imaging plates using an Echo liquid handler following the addition of 20 µL Opti-MEM media (Thermo Fisher Scientific, Cat. No. 31-985-070) using a Multidrop dispenser. The required amount of Lipofectamine RNAiMAX (Thermo Fisher Scientific, Cat. No. 13778075) was added in each well using the Echo liquid handler and incubated for 30 min at room temperature to allow siRNA-Lipofectamine complex to form. Next, the required number of cells diluted in 20 µL Opti-MEM media containing 20% FBS (Gibco, Cat. No. 10082147) was dispensed into each well. The imaging plate was incubated for 72 hrs at 37°C in a cell culture incubator. Scrambled (Thermo Fisher Scientific, Cat. No. 4390846) and all-star cell death (siDEATH) (QIAGEN, Cat. No. 1027298) siRNAs were used as controls to optimize the amount of Lipofectamine RNAiMax reagent to achieve maximum transfection efficiency with minimum cytotoxicity.

## Statistical analyses

Statistical analysis, including ANOVA, Tukey's HSD test, and t-test with Bonferroni and FDR correction, was performed using ggpubr (https://github.com/kassambara/ggpubr, *kassambara, 2026*) and rstatix (https://github.com/kassambara/rstatix/releases, *kassambara, 2025*) packages in R version 4.3.2. Statistical significance of difference was denoted by stars where * indicates $p \leq 0.05$, ** indicates $p \leq 0.01$, *** indicates $p \leq 0.001$, and **** indicates $p < 0.0001$.

## Acknowledgements

We thank members of the Misteli lab for feedback throughout this project. This research was funded by the Intramural Research Program of the NIH, NCI, Center for Cancer Research through grant 1-ZIA-BC010309-25 to TM and grant 1-ZIC-BC-011567 to HiTIF and in part with Federal funds from the National Cancer Institute, National Institute of Health under Contract No. HHSN26120150003I. Confocal imaging was performed in the CCR/LRBGE Optical Microscopy Core, funded by the Intramural Research Program of the National Cancer Institute (NCI), Center for Cancer Research (CCR): project number ZIC BC 011574, and supported by Dr. TS Karpova. This work utilized the computational resources of the NIH HPC Biowulf cluster (https://hpc.nih.gov). The content of this publication does not necessarily reflect the views or policies of the Department of Health and Human Services, nor does mention of trade names, commercial products, or organizations imply endorsement by the U.S. Government.

## Additional information

### Funding

| Funder | Grant reference number | Author |
| --- | --- | --- |
| National Institutes of Health | 1-ZIA-BC010309-25 | Tom Misteli |
| National Institute of Health Sciences | 1-ZIC-BC-011567 | Gianluca Pegoraro |
| National Institutes of Health | HHSN26120150003I | Raj Chari |

The funders had no role in study design, data collection and interpretation, or the decision to submit the work for publication.

### Author contributions

Krishnendu Guin, Conceptualization, Data curation, Formal analysis, Investigation, Writing – original draft; Adib Keikhosravi, Software, Writing – review and editing; Raj Chari, Resources, Writing – review and editing; Gianluca Pegoraro, Data curation, Formal analysis, Writing – review and editing; Tom Misteli, Conceptualization, Formal analysis, Supervision, Funding acquisition, Writing – original draft, Project administration

### Author ORCIDs

Krishnendu Guin (ID) https://orcid.org/0000-0001-6957-465X
Tom Misteli (ID) https://orcid.org/0000-0003-3530-3020

Reviewer #1 (Public review): https://doi.org/10.7554/eLife.108410.3.sa1
Reviewer #2 (Public review): https://doi.org/10.7554/eLife.108410.3.sa2
Reviewer #3 (Public review): https://doi.org/10.7554/eLife.108410.3.sa3
Author response https://doi.org/10.7554/eLife.108410.3.sa4

## Additional files

### Supplementary files

Supplementary file 1. Source, culture condition, and growth media used for cell lines used in this study.

Supplementary file 2. Results of statistical analysis for CENP-C spot count, clustering score, radial position, and nuclear area as shown in *Figure 1*.

Supplementary file 3. Sequences of sgRNAs used to target 1064 genes in the CRISPR-KO screens.

Supplementary file 4. Measurement data for CENP-C spots and nuclei obtained from CRISPR-KO screens in HCT116 Cas9 cells.

Supplementary file 5. Measurement data for CENP-C spots and nuclei obtained from CRISPR-KO

screens in hTERT-RPE1 Cas9 cells.

Supplementary file 6. Sequences of sgRNAs and DNA oligonucleotides used to generate SPC24 and NUF2 dTAG cell lines.

Supplementary file 7. Details of siRNAs used for validation of CRISPR-KO screen hits.

MDAR checklist

## Data availability

Original image files used in this article are publicly available in Figshare (https://doi.org/10.6084/m9.figshare.31224238) and analysis code used for identification of hits in the CRISPR screens is available at GitHub: https://github.com/CBIIT/mistelilab-centromeres (copy archived at *Pegoraro, 2026*).

The following dataset was generated:

| Author(s) | Year | Dataset title | Dataset URL | Database and Identifier |
|---|---|---|---|---|
| Guin K, Keikhosravi A, Chari R, Pegoraro G, Misteli T | 2026 | Original images used for preparation of figures in Guin et al., 2026, eLife | https://doi.org/10.6084/m9.figshare.31224238 | figshare, 10.6084/m9.figshare.31224238 |

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
