## [Editor Report · eLife Assessment]

This **important** study combines microscopy and CRISPR screening to identify factors involved in global chromatin organisation, using centromere clustering as a proxy. The authors present **solid** evidence demonstrating that acute depletion of a range of mitotic regulators alters centromere distribution in interphase. The work will be of interest to researchers studying genome organisation, nuclear architecture, chromosome biology, and the mechanisms linking mitosis to interphase nuclear organisation.

---

## [Referee Report · Reviewer #1 (Public review)]

Summary:

In this manuscript, Guin and colleagues establish a microscopy-based CRISPR screen to find new factors involved in global chromatin organization. As a proxy of global chromatin organization they use centromere clustering in two different cell lines. They find 52 genes whose CRISPR depletion leads to centrome clustering defects in both cell lines. Using cell cycle synchronisation, they demonstrate that centromeres-redistribution upon depletion of these hits necessitates cell cycle progression through mitosis.

Strengths:

This manuscript explores the mechanisms of global chromatin organization, which is a scale of chromatin organization which remains poorly understood. The imaging based CRISPR screeen is very elegant and use of appropriate positive and negative controls reinforces the solidity of the findings.

Weaknesses:

The manuscript shows interesting observations but left a major question unanswered: what is the functional relevance of centromeres clustering?

---

## [Referee Report · Reviewer #2 (Public review)]

The authors begin by highlighting the importance of genome organisation in cellular compartmentalisation and identity. They focus their study on centromeres - key chromosomal features required for segregation-and aim to identify proteins responsible for their spatial distribution in interphase nuclei. However, none of the experimental data addresses broader aspects of genome architecture, such as individual chromosome territories or A/B compartments. As such, the title of the article may be misleading and would benefit from being more specific, for example: "Identification of factors influencing centromere positioning in interphase."

Strengths:

One of the strengths of the paper is the comprehensive CRISPR-based screening and the comparative analysis between two distinct cell lines.

Including further investigation into factors that behave differently across these cell lines - particularly in relation to expression levels or the unique "inverted architecture" of RPE cells-would have added valuable depth.

Comments on revisions:

From the previous review:

The Authors have undertook a very minimal revision of the paper. The Authors have addressed some of the comments raised by rewarding the text and being slightly more critical in the interpretation of their results and added previously published literature.

They have provided more details on the characterisation of the new cell lines and added some statistical analyses.

However, I still believe that the title does not reflect the finding, as it is all about centromere position rather than "interphase genome architecture" as claimed.

As I said in my previous comments, this will make a precedent and will cause mis-interpretations in the field.

Changes from the previous version:

While in the new manuscript the Authors have discussed that degradation of NUF2 and SPC24 caused some aberrant nuclear phenotypes, this is at odd with the first screening where these morphologies were used as part of the exclusion criteria. Some comments would be required.

---

## [Referee Report · Reviewer #3 (Public review)]

Summary:

In this manuscript, Guin et al. use a CRISPR KO screen of ~1000 candidates in two human cell lines along with high-throughput image analysis to demonstrate that orderly progression through mitosis shapes centromere organization. They identify ~50 genes that perturb centromere clustering when depleted in both RPE1 and HCT116 cells and validate many of these hits using RNAi. They then use auxin-mediated acute depletion of four factors (NCAPH2, KI67, SPC24 and NUF2) to demonstrate that their effects on centromere clustering require passage through mitosis. They further suggest that lack of these factors during mitosis leads to disorganization of centromeres on the mitotic spindle and these effects persist in the subsequent interphase. Overall, the manuscript is clear, well-written, the experiments performed are appropriate and the data is interpreted accurately. In my opinion, the main strength of this manuscript is the discovery of several hits associated with altered centromere clustering. These hits will serve as a solid foundation for future work investigating the functional significance of centromere clustering in human cells. On the other hand, how the changes in centromere clustering relate to other aspects of interphase genome architecture (A/B compartments, chromosome territories etc) remains unclear and represents the main limitation of this manuscript.

---

## [Author Response]

The following is the authors’ response to the original reviews

**Public Reviews:**

**Reviewer #1 (Public review):**
Although the data are generally solid and well interpreted, a control showing that protein depletion works properly in cell-cycle arrested cells is lacking, both when using siRNAs and degron-based depletion.

We now demonstrate in Fig. S9 efficient degron-mediated depletion of both NUF2 and SPC24 in cell-cycle arrested cells by Western blotting. We show similar data for siRNA knockdowns. Our siRNA knockdown experiments include a “siDEATH” control that induces cytotoxicity by targeting several essential genes. In Fig. S6a we now show that siDEATH transfection results in strong cytotoxicity and cell death in cycling as well as cell cycle arrested G1/S and G2/M populations indicating efficient protein depletion. Additionally, in Fig. S6b we now show depletion NCAPH2 protein levels by siRNA knockdown in cycling as well as cell cycle arrested cell populations by Western blot analysis. We mention these results on page 11 and page 13.

**Reviewer #2 (Public review):**
The filtering strategy used in the screen imposes significant constraints, as it selects only for non-essential or functionally redundant genes. This is a critical point, as key regulators of chromatin organisation - such as components of the condensin and cohesin complexes-are typically essential for viability. Similarly, known effectors of centromere behaviour (e.g., work by the Fachinetti's lab) often lead to aneuploidy, micronuclei formation, and cell cycle arrest in G1. The implication of this selection criterion should be clearly discussed, as it fundamentally shapes the interpretation of the study's findings.

We discussed our hit selection criteria on page 8 and in the Methods section. Some of the concerns regarding a bias towards non-essential genes are alleviated by the fact that our screen is limited to a relative short duration of 72 hours rather than the longer timepoints that are generally used to assess essentiality in pooled CRISPR-KO screens, allowing us to identify genes that may be essential if eliminated permanently. In support of this notion, we identify subunits of the essential condensin and cohesin complexes as hits with only limited effect on cell viability. In this case, the Z-score for change in cell number upon NCAPH2 knockout was -0.26 indicating only a mild reduction compared to the average cell number across all targets.

Other confounding effects on hit selection due to micronuclei formation, cell cycle effects etc. are minimized as we closely monitor micronuclei formation and cell viability in our screen. Finally, aneuploidy is similarly not a confounding factor in hit identification since, as we previously demonstrated, the Ripley’s K-based clustering score is robust to changes in spot number (Keikhosravi, A., et al. 2025).

A major limitation of the study is the lack of connection between centromere clustering and its biological significance. It remains unclear whether this clustering is a meaningful proxy for higher-order genome organisation. Additionally, the study does not explore potential links to cell identity or transcriptional landscapes. Readers may struggle to grasp the broader relevance of the findings: if gene knockouts that alter centromere positioning do not affect cell viability or cell cycle progression, does this imply that centromere clustering - and by extension, interphase genome organisation - is not biologically significant?

We appreciate these points. Given the presence of one centromere on each chromosome, we used centromeres as surrogate landmarks of higher-order nuclear genome organization and considered centromere patterns as a general indicator of overall genome organization. While the relationship of centromere patterns to other genome features is poorly understood in mammalian cells, a link is suggested by observations in other organisms. For example, in yeast, the clustering of centromeres reflects the overall Rabl configuration of chromosomes. Having said that, we agree that our extrapolation to overall genome organization is somewhat speculative, and we have toned down these conclusions throughout the manuscript.

We agree that one of the most interesting questions emerging from our study is whether centromere clustering has a functional role. In follow-up studies we will use some of the key regulator identified in these screens to perturb the native centromere distribution and assay for various cellular responses including in gene expression and genome integrity. These studies will be the subject of future publications.

Another point requiring clarification is the conclusion that the four identified genes represent independent pathways regulating centromere clustering. In reality, all of these proteins localise to centromeres. For example, SPC24 and NUF2 are components of the NDC80 complex; Ki-67, a chromosome periphery protein, has been mapped to centromeres; and CAP-Hs, a subunit of the condensin II complex that during G1 promotes CENP-A deposition. Given their shared localisation, it would be informative to assess aneuploidy indices following depletion of each factor. Chromosome-specific probes could help determine whether centromere dysfunction leads to general mis-segregation or reflects distinct molecular mechanisms. Additionally, exploring whether Ki-67 mutants that affect its surfactant-like properties influence centromere clustering could provide a more mechanistic insight.

We thank the reviewer for this comment. We now clarify the relationship of these proteins to centromeres in more detail on page 12. While they all have some relationship to centromeres, as would be expected if they contributed to centromere clustering, they represent multiple distinct pathways and processes.

The observed effects on clustering are unlikely due to aneuploidy as only very limited aneuploidy is observed in our cells and because Ripley’s K measurement of centromere clustering is robust to change in chromosome copy number. Follow-up studies using live cell imaging approaches are currently in progress to address some of these mechanistic questions.

Finally, the additive effects observed mild mis-segregation effects are amplified when two proteins within the same pathway are depleted. This possibility should be considered in the interpretation of the data.

We rephrased the text on page 14 based on the reviewer’s recommendations.

**Reviewer #3 (Public review):**
Given the authors' suggestion that disorderly mitotic progression underlies the changes in centromere clustering in the subsequent interphase, I think it would be beneficial to showcase examples of disorderly mitosis in the AID samples and perhaps even quantify the misalignment on the metaphase plate.

We now include in Fig. S11 examples of disordered mitotic nuclei observed in the absence of NUF2 or SPC24.

I don't quite agree with the description that centromeres cluster into chromocenters (p4 para 2, p17 para 1, and other instances in the manuscript). To the best of my knowledge, chromocenters primarily consist of clustered pericentromeric heterochromatin, while the centromeres are studded on the chromocenter surface. This has been beautifully demonstrated in mouse cells (Guenatri et al., JCB, 2004), but it is true in other systems like flies and plants as well.

We have modified this description on page 4.

**Recommendations for the authors:**

**Reviewing Editor Comments:**
(1) Proper characterisation of the cell lines used in the manuscript. Tagged proteins have been known to affect protein levels compared to the parental cell, and where this is the case (or not), it needs to be transparently shown in the manuscript.

The cell lines to conditionally deplete NCAPH2 and KI67 have previously been published, and they have been characterized to show normal expression levels of the tagged protein (Takagi et al., 2018). We also show quantification of Western blots to compare protein level of tagged SPC24 and NUF2 to that of the untagged proteins in the parental cell line (Fig. S8e-f) and discuss these results on page 11 and page 12.

(2) Demonstration of protein depletion in the degron cell lines.

We showed efficient protein depletion in the degron cell lines (Fig. S8c and S8d). In addition, we now show in Fig. S9 depletion of SPC24 and NUF2 in cells arrested at G1/S and G2/M.

(3) The study examines centromere clustering, but not genome architecture. While it is understood that a complete investigation of genome architecture is beyond the scope of the current study, the interpretation does not match the data. The authors are suggested to pay attention to this point throughout the manuscript and consider their findings in terms of centromere clustering rather than genome architecture, including changing the title accordingly.

We have toned down our statements regarding overall genome organization throughout the manuscript. Since centromeres are a natural fiducial marker for overall genome organization and a link to overall genome organization has been suggested in some organisms such as yeast, we have retained the wording in a few select instances, including the title. We also make it clear that we do not intend to draw conclusions regarding TADs or even compartments but consider centromere patterns an indicator of overall genome organization.

**Reviewer #1 (Recommendations for the authors):**
(1) Controls of depletion by western blot in synchronized cells (siRNAs and degrons) are lacking.

We now show Western blots demonstrating efficient depletion of the target proteins in degron (Fig. S9) and siRNA treated cell-cycle arrested cells (Fig. S6b).

It would have been very nice to discuss the implications of these findings further. For example, do centromere clustering changes gene expression/repression of pericentromeric heterochromatin expression? Is centromere clustering associated with specific diseases? How is global chromatin organization affecting gene expression/genome stability, etc? Although some of these aspects are unknown, a discussion about them would have been nice.

We appreciate these interesting points. These questions are the subject of our ongoing follow up studies. We now discuss possible consequences of centromere re-organization on gene expression and genome stability on page 18.

**Reviewer #2 (Recommendations for the authors):**
Major Comments:(1) Clarify Scope and Avoid Overinterpretation(a) The study exclusively investigates centromere positioning, without addressing broader aspects of genome architecture.(b) There is no established link presented between centromere positioning and higher-order genome organisation.

We have toned down our statements regarding overall genome organization throughout the manuscript. Since centromeres are a natural fiducial marker for overall genome organization and observations in yeast suggest such a link, we have retained the wording in a few select instances. We make it clear that we do not intend to draw conclusions regarding TADs or even compartments but consider centromere patterns an indicator of overall genome organization.

(c) The exclusion criteria used in the screen should be clearly explained, including the implications of selecting only non-essential or redundant genes.

We discuss on page 8 and in the Methods section the exclusion criteria used in the screen, including the implications for identifying essential genes.

(d) The authors should discuss why the identified proteins significantly affect centromere clustering but do not impact cell cycle progression.

We now discuss this topic briefly on page 9. While some hits are expected to affect both cell-cycle progression and centromere clustering (Fig. S4c), it is not a priori expected that all hits would affect both.

(2) Supplementary Figure 1This figure appears unnecessary. The co-localisation between CENP-C and CENP-A is well established in the literature, and the scoring provided does not add essential new information.

The data was included in response to repeat questions from a centromere expert. We prefer to retain this data for completeness.

(3) Differential Hits between Cell LinesFor hits that behave differently across cell lines, expression data should be provided. Are the genes equally expressed in both cell types? What is the level of depletion achieved?

It is possible that cell-type specific hits arise due to difference in expression. Cell-type specific hits may also arise due multiple other reason including cancer vs. non-cancer origin, hTERT-immortalization, cell growth properties, variation in underlying DNA sequences of the Cas9 target loci, initial state of centromere clustering to name a few. Each of these possibilities requires additional experiments to identify the exact reason for cell-type specificity of a given factor. A full analysis of the reason for cell-type specificity is, however, beyond the scope of current study.

(4) Efficiency of Cell Cycle-Specific DegradationDegradation efficiency likely varies across cell cycle stages. The authors should provide Western blots showing the extent of protein depletion at each cell cycle block.

We provide Western blot data in Fig. S9 to demonstrate efficient knockdown of proteins in G1/S and G2/M arrested cells.

(5) Figure S6 - Validation of New Cell LinesGenotyping data for the newly generated cell lines should be included, along with Western blots using protein-specific antibodies (not just the tag), compared to the parental cell line.

We provide in Fig. S7c-d genotyping data and in Fig. S8e-f Western blot data to compare levels of tagged and untagged proteins.

(6) Figure S7 - G2/M Block EfficiencyThe G2/M block appears suboptimal after 20 hours in RO-3306, with only ~50% of cells in G2/M and just 21-27% for Ki-67, where most cells remain in S phase. This raises concerns about the interpretation of mitotic depletion effects. It is possible that cells never progressed from G1 or completed S phase without Ki-67. Prior studies (van Schaik et al., 2022; Stamatiou et al., 2024) have shown delayed and uneven replication of centromeric/pericentromeric regions upon Ki-67 depletion during S phase, which could affect the readout. Live-cell imaging would be a more robust approach to confirm mitotic status.

For KI67 after RO-3306 treatment, 73 and 67% cells were arrested at the G2/M boundary in the presence or absence of KI67, respectively (Fig. S10a-b). Upon release from G2/M arrest, the proportion of G1 cells increased from 6-13% to 28-60% in all four factors tested (Fig. S10b, and d). Please note that our results are not directly dependent on release efficiency, since we use single-cell staging (Fig. 3b) and selectively analyze only G1 populations (Fig. 5c).

We are currently working towards live cell imaging, but this requires development and characterization of additional cell lines which is beyond the scope of this study.

Statistical analyses of cell cycle phase distributions should also be included.

We include statistical analyses of cell cycle phase distributions in Fig. S4c and Fig. S10c-d by performing t-tests with FDR corrections to compare percentage of cells in either in G1, S or G2 in the presence and absence of each factor tested.

(7) Aneuploidy AssessmentAneuploidy scores for the four key proteins should be provided, ideally using centromere-specific FISH probes.

While an aneuploidy score for each hit would be interesting piece of information, we showed in a previous publication that the Ripley’s K-based Clustering Score method used here is robust to aneuploidy (Keikhosravi et al., 2025) and aneuploidy would thus not lead to spurious identification of these proteins in our screen.

(8) Add-Back Experiment (Page 14)While the add-back experiment is conceptually strong, its execution could be improved.It should be performed on synchronised cells: deplete the protein in G2/M, arrest in thymidine, then release into G1 without the protein to observe the unclustering phenotype.Re-expression should occur during the block, followed by release and analysis in the next G1 phase. This would better demonstrate whether clustering defects from the previous division can be rescued.

We have attempted these types of long-term depletion experiments in cell-cycle arrested cells, but have observed significant viability defects, making results uninterpretable.

(9) Statistical AnalysesSeveral figures lack statistical analysis, which is essential for data interpretation:(a) Figure 1B-E(b) Figure 3I(c) Figure 4B(d) Figure 5B, C, G(e) Supplementary Figures S4B and S7

Statistical analyses were performed for (a) Fig. 1b-e, (b) Fig. 3i, (c) Fig. 4b, (d) Fig. 5b-c and the details of the test are mentioned in the corresponding figure legends. We also include statistical tests for Fig. 5g, S5b and S7c-d.

Minor Comments:(1) Page 9: "Reassuringly, in line with known centromere-nucleoli association (Bury, Moodie et al. 2020, van Schaik, Manzo et al. 2022)..."The citation "van Schaik, Manzo et al. 2022" is incorrect and should be revised.

We have removed this reference.

(2) Page 10:"...were grouped into six categories: regulators of chromatin structure, kinetochore proteins, nucleolar proteins, nuclear pore complex components..."The authors should note that NUP160, listed as a nuclear pore complex hit, is also a kinetochore component during mitosis and may be linked to mitotic defects.

We now mention this on page 10.

(3) Page 12:"Progression through S phase was equally efficient in the presence or absence of KI67."While bulk S phase progression may appear unaffected, refined analyses (e.g., Repli-seq, EdU patterning) have shown delayed replication of centromeric/pericentromeric regions upon Ki-67 depletion. This should be acknowledged, especially given the study's focus on centromeres (see Schaik et al., 2022; Stamatiou et al., 2024).

Our statement was meant to describe the results we observed in this study. We indicate that overall progression is not affected, but subtle effects may persist, and we cite the relevant references on page 13.

(4) Page 12:"KI67 is a well-known marker of cell proliferation..."The first study demonstrating the dependency of chromosome periphery on Ki-67 was Booth et al., 2014, which should be cited.

This citation has been added.

**Reviewer #3 (Recommendations for the authors):**
(1) On page 14, paragraph 1, the authors suggest that NCAPH2 and SPC24 act independently on centromere clustering. I'm not convinced that this is the right interpretation of the data. Rather, the lack of an additive phenotype following NCAPH2 and SPC24 dual depletion suggests to me that these two proteins are acting in the same pathway.

We show that knockdown of NCAPH2 and SPC24 results in opposite effects in centromere clustering. However, knockdown of SPC24 in NCAPH2-AID cells produces an intermediate level of clustering compared to depletion of NCAPH2 or SPC24 knockdown alone. This indicates additive effects. We have modified our description of these results on p. 14.

(2) The analysis and experimental design in Figure 5g could be improved. For one, I would add statistical comparisons like the other figure panels. Second, the authors would ideally perform AID depletion in a synchronized G2 population before washout during the subsequent G1. This design might make some of the more subtle changes (e.g., KI67-AID) more obvious.

We now include statistical analysis in Fig. 5g. We have attempted long-term depletion experiments in cell-cycle arrested cells, but have observed significant viability defects, making results uninterpretable.

(3) In the discussion, the authors allude to centromere clustering data from the NDC80 complex, HMGA1, and other HMGs but fail to direct the reader to where they may find the data. If these data are in Tables S4 and S5, perhaps the authors could make these tables more reader-friendly?

For each target, the mean Z-score of two biological replicates based on Clustering Score is located in column H in Table S4 and S5.

(4) In my opinion, the term 'clustering score' comes across a bit ambiguous. In most cases, this term appears to refer to the distance between centromeric foci but is used occasionally to refer to the number of centromeric spots. For example, on page 9, paragraph 1, line 3, cluster/clustering is used three times but with slightly different meanings. Perhaps the authors can consider using the word 'clustering' to indicate the number of spots, 'dispersion' to indicate distance between centromeres, and 'radial distribution' to indicate distance from the nuclear center? Or other ways to improve the consistency of the descriptive terms.

We apologize for not being clear. The Clustering Score is a very specific parameter derived from use of a Ripley’s K clustering algorithm as described in Materials and Methods. We now ensure that the term is used correctly throughout and that the other terms are also used consistently.